# Molecular simulations reveal that heterogeneous ice nucleation occurs at higher temperatures in water under capillary tension

Elise Rosky[1], Will Cantrell[1], Tianshu Li[2], Issei Nakamura[1], and Raymond A. Shaw[1]

[1]Department of Physics, Michigan Technological University, Houghton, MI, USA
[2]Department of Civil & Environmental Engineering, George Washington University, Washington, DC, USA

**Correspondence:** Elise Rosky (emrosky@mtu.edu)

**Abstract.**

Heterogeneous ice nucleation is thought to be the primary pathway for the formation of ice in mixed-phase clouds, with the number of active ice-nucleating particles (INP) increasing rapidly with decreasing temperature. Here, molecular dynamics simulations of heterogeneous ice nucleation demonstrate that ice nucleation rate is also sensitive to pressure, and that negative pressure within supercooled water shifts freezing to higher temperatures. Negative pressure, or tension, occurs naturally in water capillary bridges and pores, and can also result from water agitation. Capillary bridge simulations presented in this study confirm that negative Laplace pressure within the water increases heterogeneous freezing temperatures. The increase in freezing temperature with negative pressure is approximately linear within the atmospherically relevant range of 1 to −1000 atm. An equation describing the slope depends on the latent heat of freezing and the molar volume difference between liquid water and ice. Results indicate that negative pressures of −500 atm, which correspond to nanometer-scale water surface curvatures, lead to a roughly 4 K increase in heterogeneous freezing temperature. In mixed-phase clouds, this would result in approximately one order of magnitude increase in active INP concentrations. The findings presented here indicate that any process leading to negative pressure in supercooled water may play a role in ice formation, consistent with experimental evidence of enhanced ice nucleation due to surface geometry or mechanical agitation of water droplets. This points towards the potential for dynamic processes such as contact nucleation and droplet collision/breakup to increase ice nucleation rates through pressure perturbations.

## 1 Introduction

Heterogeneous freezing of water occurs when a substrate or material in contact with water catalyzes the formation of ice. The heterogeneous ice nucleation rate coefficient, referred to in this paper as the intensive nucleation rate $j_{het}$, describes the number of ice nucleation events per unit area of substrate per unit time (m$^{-2}$ s$^{-1}$). It is commonly recognized that $j_{het}$ is temperature dependent, with probability of nucleation increasing as temperature is decreased. Meanwhile, less attention has been given to the pressure dependence of $j_{het}$, which will be the focus of this study. Specifically, we study the behavior of $j_{het}$ under negative pressure, with the hypothesis being that negative pressure in supercooled water will correspond to elevated temperatures of heterogeneous ice nucleation.

Negative pressure in water is prevalent throughout nature, occurring in tree xylem (Jacobsen et al., 2007), in water capillary bridges in soil (Seiphoori et al., 2020), and within nano-sized pores on atmospheric ice nucleating particles (David et al., 2020; Marcolli, 2020; Klumpp et al., 2023). Negative Laplace pressure, given by $\Delta P \approx \sigma_{lv}/2r$, (where $\sigma_{lv} \approx 0.7 \, \text{J m}^{-2}$ is the liquid-vapor surface tension of water), becomes significant for nucleation when the air-water interface radius of curvature $r$ is on the order of nanometers. This corresponds to negative pressures on the order of hundreds of atmospheres. Thus, in this work we explore atmospherically-relevant negative pressures down to $-1000$ atm.

Though small changes in temperature exert a much larger influence on ice nucleation rate compared to changes in pressure, our results show that negative pressures within the range of $-1000$ atm can cause heterogeneous ice nucleation rates to occur several Kelvin higher compared to without any pressure change. Within the narrow band of temperatures where heterogeneous ice nucleation is active in the atmosphere, it is commonly approximated that the concentration of active ice-nucleating particles increases exponentially with decreasing temperature (Pruppacher and Klett, 1997, Sec. 9.2); this suggests the activation of ice-nucleating particles a few Kelvin warmer can have significant impacts.

Recent experiments provide compelling evidence that dynamic or geometric factors can lead to enhancement of ice nucleation rates independent of temperature, leading investigators to explore non-thermal sources of freezing enhancement. For example, it is widely known that contact between an ice-nucleating material and supercooled liquid leads to an increase in the freezing temperature as compared to when the material is immersed (Pitter and Pruppacher, 1973; Levin and Yankofsky, 1983; Diehl et al., 2002). Wetting of a small ice-nucleating particle at a water surface (Shaw et al., 2005) and roughening of a substrate (Gurganus et al., 2014) have both yielded similar increases. Even contact with a soluble material not typically considered an ice nucleating particle can induce freezing (Niehaus and Cantrell, 2015). It has been observed that ice nucleation is strongly enhanced when the three-phase contact line of a sessile drop distorts and moves over a substrate during electrowetting (Yang et al., 2015) or over a surface with pinning points (Yang et al., 2018). One hypothesis that attempts to unify these diverse observations is that the curvature and/or stretching of the air-water interface produces negative Laplace pressure and tension within the water (Marcolli, 2017; Yang et al., 2020).

Our previous molecular dynamics simulations of homogeneous ice nucleation within pure water identified that homogeneous ice nucleation rates occur at higher temperatures due to negative pressure, where the increase in temperature $\Delta T$ resulting from a decrease in pressure $\Delta P$ is described by the linear approximation (Rosky et al., 2022)

$$\Delta T = \frac{T_m \Delta \nu_{ls}}{l_f} \Delta P. \tag{1}$$

The governing quantities are the equilibrium melting point ($T_m$), the molar-volume difference between liquid and solid water ($\Delta \nu_{ls} = \nu_l - \nu_s$), and the enthalpy of fusion (also known as latent heat, $l_f$) — all evaluated at a reference pressure of 1 atmosphere. The molar volume difference between liquid water and ice is negative across the range of pressures addressed in this study. This property of water allows for an increase in freezing temperature ($\Delta T > 0$) from a decrease in pressure ($\Delta P < 0$). A thorough derivation of Eq. (1) can be found in the appendix of Rosky et al. (2022), using the pressure-dependent formulation of the solid-liquid chemical potential difference ($\mu$) formulated by Němec (2013) combined with classical nucleation theory

(see also Yang et al., 2018). Several thermodynamic properties of water such as the ice and liquid density, latent heat of fusion, and ice–liquid surface tension are approximated as constant with temperature and pressure on the scales relevant to this work.

In Equation 1, the slope of $(\Delta T/\Delta P)_{hom}$ is approximated as parallel to the liquid-solid phase coexistence (melting point) line, given by the Clapeyron equation. However, it is important to note that homogeneous freezing lines are not actually parallel to the melting point line as shown in both experiments and simulations of water (Bianco et al., 2021; Espinosa et al., 2016; Kanno et al., 1975; Lu et al., 2016; Dhabal et al., 2022). The slope of $(\Delta T/\Delta P)_{hom}$ becomes increasingly steeper than the melting point line at large positive pressures. However, all of these studies suggest that the homogeneous freezing lines and melting point line become closer in slope when approaching the negative pressure regime. Indeed, Rosky et al. (2022) has shown that approximating the two slopes as parallel is satisfactory at negative pressures ranging from 1 atm to $-1000$ atm. Furthermore, inspection of the simulated thermodynamic properties of water by Montero de Hijes et al. (2023) arrived at a similar conclusion that "the homogeneous nucleation line should be parallel to the coexistence line for pressures below 500 ∼bars." In light of this interpretation, we see that applying Eq. (1) at moderate positive pressures (e.g. 200 atm) provides excellent agreement with the experimental homogeneous freezing temperature depression measured by Kanno et al. (1975), whereas extending the approximation of Eq. (1) farther into the positive pressure regime (e.g., beyond 500 atm) exhibits a growing discrepancy between the predicted value and the experimental measurement. Given the lack of experimental measurements of ice nucleation at negative pressures, it is helpful to ensure that the water models used for this study compare well with measurements in the positive regime before extrapolating to negative pressures. We describe in the Methods section how the water models employed in this study show reasonable agreement with experimental measurements of homogeneous freezing at positive pressures, which builds confidence that they can be used to derive useful insights pertaining to freezing of real water under negative pressures.

In the present study we ask: Does the heterogeneous ice nucleation rate follow a similar expression for pressure dependence at negative pressures as that of the homogeneous ice nucleation rate? And do simple geometric arrangements leading to negative Laplace pressure indeed result in similar behavior? The behavior of homogeneous ice nucleation at negative pressures has been the subject of investigation because of its relationship to the fundamental properties of water (e.g., Bianco et al., 2021; Lu et al., 2016). The work of Rosky et al. (2022) used molecular simulations to characterize a simple (linear) behavior of homogeneous ice nucleation rate across the range of negative pressures thought to be relevant to the atmosphere. Extending these studies to heterogeneous ice nucleation is an integral step towards applying these findings to physical situations. In particular, heterogeneous ice nucleation in atmospheric cloud droplets is of great interest because the majority of ice in the atmosphere forms via this mechanism (Cantrell and Heymsfield, 2005; Hoose and Möhler, 2012). Heterogeneous ice nucleation rates determine the temperature at which primary ice particles form in clouds, which goes on to influence the cloud optical properties, lightning activity, and precipitation (Lamb and Verlinde, 2011).

In this work we characterize the pressure dependence of intensive heterogeneous ice nucleation rates $j_{het}$ at negative pressures using a molecular model of water in contact with an ice-nucleating substrate. In the first method of applying negative pressure we use a barostat to explicitly set the pressure. In the second method we use capillary water bridges of varying heights to create a range of negative Laplace pressures. We consider whether Eq. (1) remains a valid description for the pressure de-

pendence of heterogeneous ice nucleation in the negative pressure range. In addition to analyzing heterogeneous ice nucleation rates, we also explore the spatial distribution of ice nucleation events within the water capillary bridges to understand how confinement relative to the substrate and the air-water interface may influence our results.

While our simulations do not attempt to represent any specific substrate or configuration found in atmospheric or experimental ice nucleation, our findings provide insight into the extent that capillary tension and surface curvature (e.g. due to mechanical agitation) can influence heterogeneous ice nucleation rates through negative Laplace pressure. Additionally, the spatial locations of ice nucleation relative to the substrate and to the air-water interface inform us that these length scales must be taken into account when considering ice nucleation rate enhancement via this mechanism.

## 2 Methods

Molecular dynamics (MD) simulations are carried out in LAMMPS (Plimpton, 1995) using the mW (Molinero and Moore, 2009) water model, and the ML-mW (Machine-Learning-mW) model which has a more realistic representation of the density difference between liquid water and ice (Chan et al., 2019). The mW homogeneous freezing curve was explored across a wide range of positive pressures by Lu et al. (2016), showing qualitative agreement with that measured by Kanno et al. (1975). More relevant to our study, the ML-mW model also exhibits quantitative agreement with the experimental measurements. Using the same methods described in Rosky et al. (2022), we simulated homogeneous freezing of the ML-mW model at a positive pressure of 500 atm and observed a 6 K depression in the freezing temperature in reference to 1 atm. This agrees quantitatively with the experimental results of Kanno et al. (1975), and exhibits the expected steepening of the homogeneous freezing line in the positive pressure regime. This increases our confidence that ML-mW simulation results will provide useful insights pertaining to real water at negative pressures.

The selection of pressures used for this study is informed by two factors. First, Eq. (1) dictates that an increase in freezing temperature due to negative pressure is expected when the molar volume difference between liquid water and ice is negative in value ($\Delta\nu_{ls} < 0$). Simulations of water under a wide range of thermodynamic conditions indicate that $\Delta\nu_{ls}$ changes from negatively to positively valued at some point below $-1000$ atm (Bianco et al., 2021). Thus, we do not explore pressures below $-1000$ atm to stay within the range of interest to ice nucleation enhancement. Furthermore, we are particularly interested in negative pressure regimes that could be feasible during atmospheric processes or during laboratory experiments, making the range of pressure from 1 atm to $-1000$ atm appropriate for our purposes.

### 2.1 Ice nucleation rate

To observe ice nucleation, we equilibrate the water at a supercooled temperature and then cool the water at a constant rate of $0.25$ K ns$^{-1}$ until ice forms. The same cooling rate is used for all simulations in this study. All simulations employ periodic boundary conditions along the three spatial dimensions, and use a simulation timestep of 5 fs. The starting temperature for the cooling ramp at all pressures is 240 K for ML-mW and 225 K for mW, corresponding to 52 K and 48 K of supercooling respectively, relative to their melting temperatures at 1 atm. Ice is identified using the $q_6$ order parameter, where clusters of

125 molecules with $q_6 > 0.54$ are considered to be ice (Steinhardt et al., 1983; Lupi et al., 2014; Rosky et al., 2022). Using the same method as Rosky et al. (2022), the freezing temperature of each cooling ramp is identified by the sharp increase in ice to liquid ratio. The steepest point on a sigmoidal fit to the ice/liquid ratio curve is used to select the nominal nucleation temperature.

Intensive nucleation rates are found by running the same cooling simulation a minimum of 20 times and recording the freezing temperature of each cooling run. These repeated cooling runs form a statistical distribution of freezing temperatures

that can be divided into temperature bins with a corresponding intensive nucleation rate, $j_{het}(T)$, within each bin. Calculation of intensive nucleation rate values for this process is described in Rosky et al. (2022) using the methods of Zobrist et al. (2007). The width of the temperature bins are displayed as uncertainty bounds on our data points. As determined from Poisson statistics, we report with 99% certainty that the intensive nucleation rates shown are contained within the bounds of the temperature bin. We reference Table 2 of Koop et al. (1997) to obtain the 99% upper and lower confidence bounds for the number of freezing

events observed in each temperature bin. The cooling rate and substrate surface area used in our study allow us to sample intensive heterogeneous nucleation rates of order of magnitude $j_{het} = 10^{24}$ m$^{-2}$ s$^{-1}$.

## 2.2  Ice-nucleating substrate immersed in water

We simulate heterogeneous ice nucleation by inserting an ice-nucleating substrate into a box of water with periodic boundary conditions. This configuration is shown in Fig. 1(a). The Fig. 1(a) simulation box has dimensions of 49.22×49.02×∼58.75

Å, containing 4,906 water molecules with a 10 Å thick sheet of ice-nucleating substrate inserted at the center of the z-axis. The total surface area of substrate in contact with water (including both the top and bottom of the substrate) is 48.25 nm$^2$. The substrate molecules are held fixed with zero velocity. The x,y-dimensions of the simulation cell are kept constant, while the z-axis dimension may adjust to accommodate changes in pressure. After equilibration of the simulation box, we use cooling ramps to measure $j_{het}(T)$ at three pressures: 1 atm, −500 atm, and −1000 atm.

Simulations containing 4,906 molecules (a volume of roughly 125 nm$^3$) are a conventional choice for studies that aim to simulate bulk water properties (e.g., Molinero and Moore, 2009; Lupi et al., 2014; Li et al., 2011). The interaction forces felt between two water molecules in our simulation go to zero when molecules are spaced further than ∼4 Å apart, and forces between water molecules and substrate molecules go to zero for separations beyond ∼5 Å (Molinero and Moore, 2009). Thus, the chosen thickness of our substrate ensures that water molecules on either side of the substrate will not interact with each

other. We refer to the configuration shown in Fig. 1(a) as "unconfined" because the vertical dimension is large enough that much of the water is unaware of the substrate. Structural correlations in the liquid decay to zero beyond 1.5 nm (Cox et al., 2015; Bi et al., 2016; Lupi et al., 2014); the height of the "unconfined" simulation cell is more than twice this length. To confirm that the nucleation rate on the substrate is not influenced by confinement effects in the "unconfined" cell, simulations at 1 atm were performed with the $z$-axis dimension doubled to 100 Å, showing no change in the heterogeneous nucleation rate.

## 2.3  Confinement

To simulate the effects of confining the water between the two substrate surfaces, we repeat the simulations with reduced box heights, as shown in Fig. 1(b). We test $z$-axis heights of $h = 30$ Å, 24 Å, and 18 Å, which will later be used as the heights of our

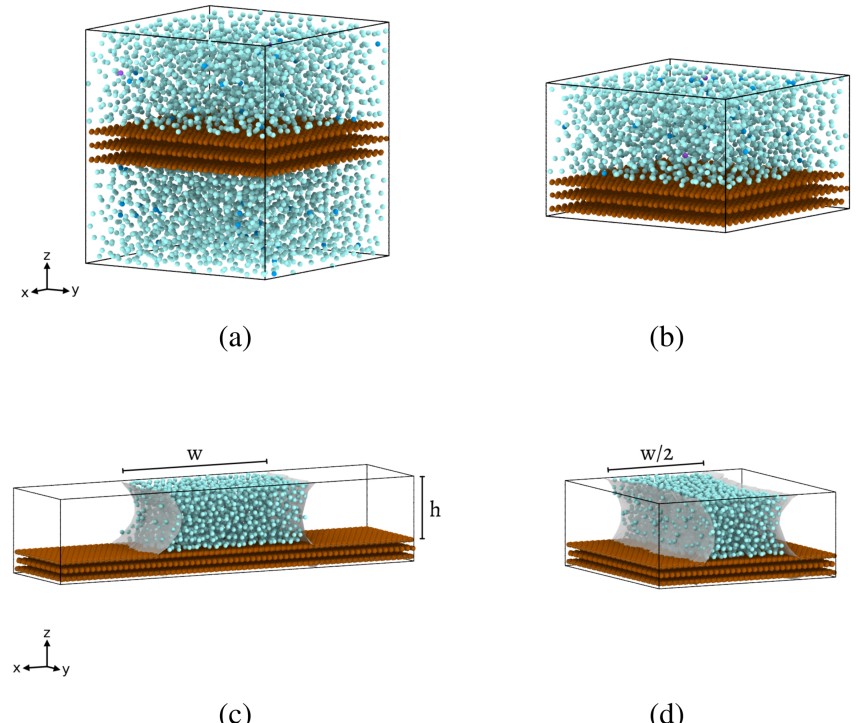

**Figure 1.** Heterogeneous ice nucleation simulation configurations with periodic boundary conditions employed along all three dimensions. Water molecules are cyan and the substrate material is brown. All configurations have the same total surface area of contact between water and substrate to within 6%. (a) "Unconfined" water with a barostat. (b) Confined water with a barostat, designed to capture the effect of confining water along the $z$-axis between substrate surfaces. (c) Capillary water bridge, used without a barostat so that negative Laplace pressure may arise naturally within the water capillary bridge due to the curved geometry of the air-water interface (gray shading). (d) Narrow capillary bridge, simulates confinement of the water along the $x$-axis between the two air-water interfaces (gray shading). In configurations (b), (c), and (d), the periodic boundary conditions cause the bottom surface of the substrate to be in contact with the water molecules at the top of the simulation cell, thus forming the water bridge in (c) and (d).

capillary water bridges. For both the "unconfined" and confined configurations, equilibration is carried out with a Berendsen barostat (damping constant of 10 ps) and Bussi thermostat (damping constant of 5 ps). The cooling runs are carried out in the isothermal-isobaric (NPT) ensemble, implemented by employing an isobaric-isoenthalpic (NPH) ensemble using a Nose-Hoover barostat (damping constant of 10 ps) coupled with the Bussi thermostat (Allen and Tildesley, 2017). Only the ML-mW model is used for the confinement simulations.

## 2.4 Water capillary bridges

We have taken steps towards addressing whether these magnitudes of negative pressure within water can be found in nature by simulating water capillary bridges. A volume of water placed between two hydrophilic substrate surfaces forms a capillary

bridge that is expected to have negative Laplace pressure within the water. As shown in Fig. 1(c), we use the same ice-nucleating substrate used in the previously described configurations to construct water capillary bridges with heights $h = 30$ Å, 24 Å, and 18 Å. We apply the same constant-cooling simulation procedure to obtain intensive heterogeneous ice nucleation rates within the capillary bridges. In these simulations, we remove the external barostat so that negative pressure within the water is solely a result of the capillary bridge geometry. Only the ML-mW model is used for the capillary bridge simulations.

Water capillary bridge simulations are equilibrated and carried out in the NVT ensemble (number of molecules, volume, and temperature is conserved during each timestep), using a Bussi thermostat with damping constant of 5 ps. The simulation cell dimensions are all held fixed during the capillary bridge simulations. The widths of these capillary bridges are measured at the water–substrate interface. The total area of contact between the water and substrate remains consistent ($\sim 48.25$ nm$^2$) to within 6% for all capillary bridge configurations. The desired contact area is achieved by tuning the y-axis dimension of the simulation cells to adjust the length of the capillaries along the y-axis. Intensive nucleation rates are ultimately obtained by dividing by the water–substrate surface area (m$^{-2}$ s$^{-1}$) and maintaining a consistent surface area ensures that all simulations sample the same magnitude of nucleation rate.

We look more closely at the ice nucleation events within the water capillary bridges by identifying the location of all nucleation events. The identification of ice nucleation location is done visually using the clustering capability of OVITO, a visualization and analysis tool for molecular dynamics simulation data (Stukowski, 2010). The center of mass of the initial ice cluster is used as the freezing location. The initial ice clusters contain an average of 25 water molecules and their positions have an uncertainty of 5 Å along each axis.

Throughout this paper, data points will be presented using circles to indicate "unconfined" heterogeneous freezing in the configuration shown by Fig. 1(a). Squares will indicate confined heterogeneous freezing as in the example of Fig. 1(b). Diamonds will indicate water capillary bridge data, as in Fig. 1(c). Narrow diamonds will represent the narrow capillary configuration, shown by Fig. 1(d).

## 2.5   Interaction potential between water and substrate

In molecular dynamics simulation, the forces between molecules are defined by interaction potentials. Lupi et al. (2014) introduced an interaction potential to model mW water with a carbon substrate, with the contact angle between mW water and carbon tuned to 86 degrees. We use this same interaction potential to simulate heterogeneous freezing of the mW water model. Although our focus in this work is not to model any particular substrate, this mW–Carbon interaction potential has the benefit of having been already used in studies of ice nucleation (Lupi et al., 2014; Bi et al., 2016). A description of the equations of interaction for this potential and its parameters is included in Appendix A.

We observe that a simulation box containing 4,096 mW water molecules at 1 atm with no substrate freezes homogeneously at an average temperature of 202 K when cooled at a constant rate of 0.25 K ns$^{-1}$. After inserting a layer of substrate into the center of this same volume of water (as in Fig. 1(a)) and using the same cooling rate, we observe that heterogeneous freezing on the substrate takes place at an average temperature of 217.5 K, a 15.5 K increase over the homogeneous freezing

temperature. This is consistent with the results of Lupi et al. (2014), who reported a $12\pm3$ K difference between heterogeneous and homogeneous freezing temperature when cooling at a rate of 1 K ns$^{-1}$.

Our previous study (Rosky et al., 2022) indicates that the ML-mW water model is more appropriate for studying pressure effects on ice nucleation because it exhibits a molar volume difference between liquid water and ice that is closer to real water (Chan et al., 2019). Therefore, in order to increase the ability to translate our findings onto real water droplets, we focus our study on the ML-mW model instead of mW. We define a substrate interaction potential for the ML-mW model that is a modified version of the mW–Carbon interaction potential defined by Lupi et al. (2014). We adjust the parameters of the interaction potential to form a substrate on which ML-mW water is stable within the range of negative pressures that we study and exhibits a similar temperature difference between heterogeneous and homogeneous freezing as seen in the mW–Carbon interaction. A simulation box containing 4,096 ML-mW water molecules at 1 atm freezes homogeneously at an average temperature of 214.5 K when cooled at a constant rate of 0.25 K ns$^{-1}$. With the substrate inserted, the average heterogeneous freezing temperature on this substrate is 228.5 K, a change in temperature of 14 K.

The resulting ML-mW–Substrate interaction potential has a contact angle between water and substrate that is smaller than that of the mW–Carbon potential. This smaller contact angle is consistent with Bi et al. (2016), Cox et al. (2015) and Lupi and Molinero (2014), where the interaction potential is adjusted in a similar manner as here to modify the substrates hydrophilicity. Our estimate of the ML-mW–Substrate contact angle is 50.5 degrees and will be discussed in Sec. 3.2. The resulting interaction parameters used in this work are summarized in Table A1.

## 3 Results

### 3.1 Heterogeneous nucleation rate with negative pressure

Freezing of water on an ice-nucleating substrate (Fig. 1(a)) is simulated at 1 atm, $-500$ atm and $-1000$ atm to identify how the intensive heterogeneous nucleation rate, $j_{het}$, behaves at negative pressures. The cooling rate and area of the substrate in contact with water is kept fixed as we change the pressure of the system so that the same magnitudes of $j_{het}$ are sampled at all pressures. For each pressure setting, we identify the temperature at which the intensive nucleation rate $j_{het}$ is equal to $10^{24}$ m$^{-2}$s$^{-1}$, thus obtaining contours of constant $j_{het}$ in pressure–temperature coordinates. Figure 2 shows these results for both the ML-mW model and mW model. Intensive homogeneous nucleation rate data $j_{hom}$ (m$^{-3}$s$^{-1}$), as well as equilibrium melting points $T_m$ are also included in these plots for comparison with Rosky et al. (2022).

Comparing the data for intensive heterogeneous nucleation rate ($j_{het} = 10^{24}$ m$^{-2}$s$^{-1}$) with the data for intensive homogeneous nucleation rate ($j_{hom} = 10^{32}$ m$^{-3}$s$^{-1}$), we see that the two follow a similar slope in pressure–temperature coordinates. Most significantly, we observe that the increase in temperature as a function of pressure for $j_{het}$ can be approximated as linear within the sampling uncertainty, indicating that the use of a linear estimate for $(\Delta T/\Delta P)_{het}$ may be applied to heterogeneous ice nucleation. For the mW model in particular, the slope predicted by Eq. (1) fits exceptionally well to both the homogeneous and heterogeneous freezing data. Meanwhile, Eq. (1) seems to underestimate the heterogeneous slope of the ML-mW model. The source of excess steepness in the ML-mW heterogeneous slope is not uncovered in this study and merits further investiga-

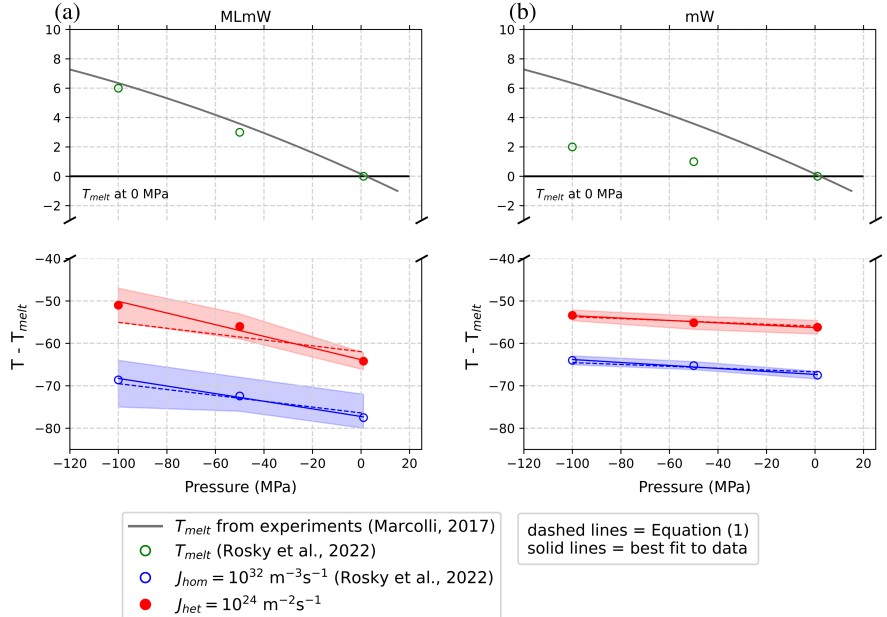

**Figure 2.** Melting (green open circles), heterogeneous-freezing (red closed circles) and homogeneous freezing temperature (blue open circles) versus pressure. The heterogeneous freezing results are new, using an ice-nucleating substrate inserted into a filled box of water with periodic boundary conditions (See Fig. 1(a)). The melting and homogeneous freezing results are reproduced from Rosky et al. (2022) for comparison. The red and blue shading represents the 99% confidence intervals (uncertainty) for the simulation data. Dashed lines use the slope predicted by Eq. (1) to obtain a best fit to the intercept. Solid lines are a best linear fit of both slope and intercept. The numeric values of these slopes are given in Table 1. Contours of constant heterogeneous nucleation rate are, to within sampling uncertainty, linear and nearly parallel with lines of constant homogeneous nucleation rate for both the ML-mW model (a) and mW model (b). The gray melting point line from Marcolli (2017) is a fit to experimental measurements, which the ML-mW model reproduces more realistically. Note that $-100$ MPa $= -1000$ atm.

tion. Despite this weakness, these results do indicate that the slope predicted by Eq. (1) may still be applicable to heterogeneous nucleation, to within the simulation uncertainty. The similarity between the heterogeneous and homogeneous slopes for each respective water model suggest that $l_f$ and $\Delta v_{ls}$ remain key factors in determining the pressure dependence of heterogeneous nucleation rate. Indeed, the consistent change in slope between the mW and ML-mW models indicate that the magnitude of density difference between liquid water and ice plays a similar role for heterogeneous freezing as was previously found for homogeneous freezing (Rosky et al., 2022).

In classical nucleation theory, $j_{het}$ has the form (Lamb and Verlinde, 2011)

$$j_{het} = A_{het} \exp\left(\frac{C f_{het}}{T \Delta \mu^2}\right), \tag{2}$$

where $f_{het}$ is a heterogeneous compatibility function, typically related to the contact angle of water on the substrate, and $\Delta\mu$ is the chemical potential difference for the phase change. The factor $C = 16\pi\gamma_{ls}^3/(3k_B\rho^2)$ depends on the liquid-solid surface tension ($\gamma_{ls}$) and density of ice ($\rho$). The pre-factor $A_{het}$ is related to the diffusivity of water molecules. The pressure dependence is introduced into this expression by using a formulation for chemical potential difference given by Němec (2013). As with homogeneous freezing, we may approximate that the pre-factor, liquid and ice density, latent heat of fusion, and ice–liquid surface tension remain constant with temperature and pressure on the scales relevant to this work. As long as the heterogeneous compatibility function $f_{het}$ is not strongly pressure dependent, the approximation of Eq. (1) is valid for heterogeneous, as well as homogeneous, ice nucleation. The heterogeneous compatibility function, $f_{het}$ of the substrate, is formulated as a function of the effective contact angle between water and substrate (Lamb and Verlinde, 2011; Zobrist et al., 2007)). The roughly linear trend in our data suggest that the contact angle and compatibility function are not strongly dependent on pressure in the negative pressure regime studied here. This follows from the fact that the derivation of Eq. (1) hinges on approximating many terms as constant along lines of constant $j$; if these assumptions are invalid in the pressure regime examined here, we would expect their effects to reveal themselves by producing a highly non-linear trend in pressure–temperature coordinates, which we do not observe.

The values of $T_m$, $\Delta\nu_{ls}$, and $l_f$ used in Eq. (1) to produce the dashed lines in Fig. 2 are listed in Table 1 and correspond to bulk water at 1 atm (no proximity to an interface). As summarized in column four of Table 1, these values predict a slope of $(\Delta T/\Delta P)_{het} = -0.069$ K MPa$^{-1}$ for the ML-mW model. A line of best fit to the simulated ML-mW heterogeneous freezing data instead gives a slope of $(\Delta T/\Delta P)_{het} = -0.14$ K MPa$^{-1}$ (solid red line in Fig. 2(a)). The predicted slope (dashed red line in Fig. 2(a)) sits only marginally within the uncertainty bounds of the heterogeneous nucleation rate data and is 47% steeper than the best fit of the ML-mW homogeneous freezing data (See Table 1).

We take some time to consider which factors may contribute to the steeper slope of heterogeneous freezing in the ML-mW model compared to the homogeneous freezing line. One hypothesis is that Eq. (1) still holds true for heterogeneous ice nucleation, but that adjustments need to be made to the values of $\Delta v_{ls}$ or $l_f$ used in the equation to achieve quantitative agreement with the heterogeneous freezing line. While our results support that the trend in $(\Delta T/\Delta P)_{het}$ can be approximated as linear in the negative pressure regime, we are less confident in the quantitative values of $\Delta v_{ls}$ and $l_f$ for heterogeneous freezing compared to homogeneous freezing. In the homogeneous case, the values of $T_m$, $l_f$, and $\Delta v_{ls}$ are well constrained because they can be calculated from bulk water, which could account for the excellent agreement between Eq. (1) and the homogeneous simulation results of Rosky et al. (2022). For the heterogeneous case, there is more ambiguity around these thermodynamic values because the properties of water near an interface may differ from the bulk properties. We see that by using bulk thermodynamic values in Eq. (1), we underestimate the slope of $(\Delta T/\Delta P)_{het}$ by up to 50%. Interestingly, this discrepancy between the homogeneous and heterogeneous slopes is seen only in the ML-mW model. Meanwhile the homogeneous and heterogeneous freezing lines are nearly parallel for the mW model. This could indicate the thermodynamic properties of mW water are less influenced near the substrate compared to the ML-mW model. This interpretation is supported by evidence from Qiu et al. (2018) which identified that mW water at the mW–Carbon interface has thermodynamics similar to that of the bulk liquid.

**Table 1.** Parameters entering Eq. (1) for homogeneous and heterogeneous nucleation, for both mW and ML-mW models. Dashes indicate that a measurement of this value is not known to us. Sources: [a] Chan et al. (2019), [b] Rosky et al. (2022).

| | $l_f$ | $T_m$ | $\Delta v_{ls}$ | $\frac{T_m \Delta v_{ls}}{l_f}$ | $\Delta T/\Delta P$ best fit |
| --- | --- | --- | --- | --- | --- |
| | [J/mol] | [K] | [m³/mol] | [K/MPa] | [K/MPa] |
| mW $J_{hom}$ | 5271.8 [a] | 273 [a,b] | $-4.2 \times 10^{-7}$ [a,b] | $-0.022$ | $-0.035$ |
| mW $J_{het}$ | - - - | 273 | - - - | | $-0.028$ |
| ML-mW $J_{hom}$ | 5857.6 [a] | 292 [b] | $-13.8 \times 10^{-7}$ [a,b] | $-0.069$ | $-0.089$ |
| ML-mW $J_{het}$ | - - - | 292 | - - - | | $-0.14$ |
| Real water $J_{hom}$ | 6025.0 [a] | 273.15 [a] | $-16.1 \times 10^{-7}$ [a] | $-0.073$ | - - - |

Another possibility is that the contact angle and surface free energy (surface tension) between water and substrate exerts an influence on the slope of $(\Delta T/\Delta P)_{het}$. Evans (1967) showed that a heterogeneous freezing line for aqueous suspension of ice-nucleating material at positive pressures is actually less steep than the melting point line, and thus also less steep than the homogeneous freezing line. Although the mW–substrate and ML-mW–substrate were designed to have the same freezing temperature enhancement over the homogeneous freezing temperatures, they do exhibit different contact angles between the water and substrate. Thus, this could potentially be a factor involved in the discrepancy between ML-mW homogeneous and heterogeneous slopes as compared to the mW case where the two lines are parallel. Note that in the previously mentioned study by Qiu et al. (2018), the bulk-like thermodynamics of mW at the substrate interface can be attributed to the nearly 90 degree contact angle between mW and Carbon.

In our simulation results, using bulk water thermodynamic values in Eq. (1) provides a lower-bound to the slope of $(\Delta T/\Delta P)_{het}$, which can be very useful in estimating the increase in heterogeneous freezing temperature due to negative pressure in atmospheric and experimental contexts. However, further investigation is needed to identify a robust way to account for the observed steepness of the ML-mW heterogeneous nucleation slope. Going forward, the linearity and the theoretical prediction of the slope are probably adequate for practical use, given other significant uncertainties in using classical nucleation theory.

### 3.2 Heterogeneous ice nucleation in water capillary bridges

One way for water to exist stably under negative pressure is through geometric configurations which produce high degrees of negative surface curvature at the air-water interface, such as inside a water capillary bridge. The negative pressure experienced by the water in these cases is a result of Laplace pressure, $\Delta P = \sigma_{lv}(\frac{1}{r_1} + \frac{1}{r_2})$, where $\sigma_{lv}$ is the surface tension between liquid and vapor, and $r_1$ and $r_2$ are the radii of curvature of the air-water interface. Negatively valued radii of curvature cause the equilibrium pressure within the water to be smaller than the external environmental pressure, allowing for negative pressure to exist within water that is otherwise in a 1 atm environment. The Laplace pressure associated with different capillary geometries is summarized by Elliott (2021). For the capillary bridge configuration used in this study, shown in Fig. 1(c), the expected Laplace pressure within the water is

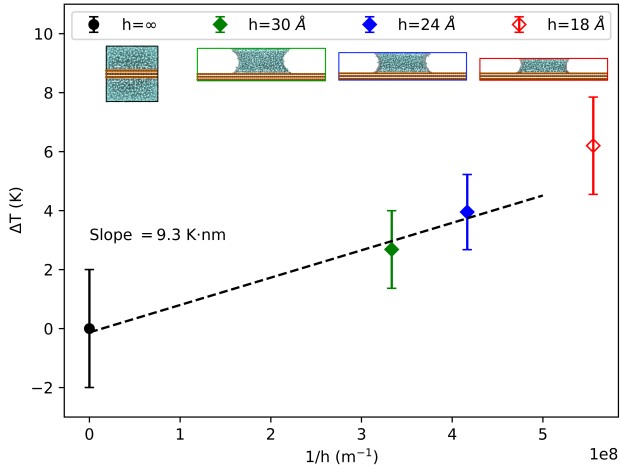

**Figure 3.** Increase in heterogeneous freezing temperature $\Delta T$ versus $1/h$ for capillary-bridge heights of $h = 30$ (green filled diamond), 24 (blue filled diamond), and 18 Å (red open diamond), as well as for bulk water (filled black circle) in contact with identical ice-nucleating substrates. Excluding the 18-Å capillary (red diamond) which is influenced by confinement effects, the line of constant nucleation rate as a function of inverse capillary height follows a linear trend shown by the linear best fit to the data (dashed black line). The 18-Å capillary data is not included in the linear fit to obtain the indicated slope $\Delta T \cdot h$.

$$\Delta P = -\sigma_{lv} \frac{2\cos(\theta)}{h}, \tag{3}$$

where $\theta$ is the contact angle between water and the substrate. By substituting the above equation into Eq. (1), we obtain an expression to predict the temperature increase for a given nucleation rate $j_{het}$ as a function of inverse capillary bridge height,

$$\Delta T = -2\sigma_{lv} \cos(\theta) \frac{\Delta\nu_{ls}T_m}{l_f} \left(\frac{1}{h}\right). \tag{4}$$

Given the previous inference that terms $\sigma_{lv}$ and $\theta$ do not change significantly with pressure, we expect a linear relationship between freezing temperature and inverse capillary height $1/h$. Figure 3 shows the freezing temperatures corresponding to an intensive heterogeneous nucleation rate $j_{het} = 10^{24}$ m$^{-2}$s$^{-1}$ inside water capillary bridges with heights $h = 30$, 24, and 18 Å. We find that the data can be described by a linear trend as anticipated. As will be discussed in Sec. 3.4, we have excluded the 18 Å capillary bridge from the current analysis because this scale of confinement of the water between the substrate surfaces causes an increase in ice nucleation rate that cannot be attributed to negative pressure alone.

We can now use the linear slope $-2\sigma_{lv}\cos(\theta)\left[\frac{\Delta\nu_{ls}T_m}{l_f}\right]$ from Eq. (4) to analyze the results in Fig. 3. A linear fit to the data, shown by the dashed black line in Fig. 3, produces a slope of $(\Delta T \cdot h) = 9.3$ K·m. We also know from our analysis of $(\Delta T/\Delta P)_{het}$ in Sec. 3.1 and Table 1 that the best fit value of $\left[\frac{\Delta\nu_{ls}T_m}{l_f}\right]$ for the ML-mW model is $-0.14$ K MPa$^{-1}$. We substitute these two slopes into Eq. (4) to solve for the value $\sigma_{lv}\cos(\theta) = 0.042$ J m$^{-2}$. The surface tension $\sigma_{lv}$ of the

mW model was reported by Molinero and Moore (2009) to be $0.066$ J m$^{-2}$, and by following methods of Li et al. (2009, Supplementary material), we found the same value for the ML-mW model. Using this value of $\sigma_{lv}$, we solve for the contact angle $\theta = 50.5$ degrees. This value of $\theta$ is consistent with estimates we obtain by measuring the radius of curvature of the capillary bridge air-water interfaces using a method similar to Giovambattista et al. (2007). With these estimates of $\sigma_{lv}$ and $\theta$, we can also use Eq. (3) to calculate the magnitude of Laplace pressure that may be present within the water capillary bridges. The 24-Å capillary bridge has a pressure of $-345$ atm, and the 30-Å bridge a pressure of $-275$ atm.

Our main findings are that negative Laplace pressure created within water capillary bridges increases the temperature of $j_{het}$ in a manner that is consistent with the linear slope of $(\Delta T / \Delta P)_{het}$ combined with the expected Laplace pressure for this capillary geometry. A 24-Å capillary bridge can exhibit a $\approx$ 3-K increase in heterogeneous freezing temperature, and a $\approx$ 2-K increase within a 30-Å water capillary bridge.

### 3.3 Effect of confinement between substrate layers

Capillary theory is expected to remain valid at the nano-scale (Elliott, 2021), however the ice nucleation rate in water can be affected by confined geometries on these scales (Cao et al., 2019; Roudsari et al., 2022; Hussain and Haji-Akbari, 2021). When confined between two flat surfaces (sometimes referred to as a "slit pore"), density oscillations in the water induced by the flat interfaces can interfere constructively and influence the ice-forming probability (Cox et al., 2015; Bi et al., 2016; Lupi et al., 2014). When analyzing the increase in freezing temperature within the water capillary bridges, we need to disentangle the effects of confinement from the effects of Laplace pressure. To do so, we ran simulations to observe how confinement alone affects ice nucleation rate on the substrate. We use the configuration shown in Fig. 1(b), where the spacing between the substrate surfaces along the $z$-axis dimension has been reduced to 18, 24 and 30 Å to match the levels of confinement present in our water capillary bridges. In these simulations, water molecules are confined between the substrate surfaces, but have no air-water interface that gives rise to Laplace pressure. The pressure in the boxes is set to 1 , $-500$, and $-1000$ atm, to directly compare against the "unconfined" configuration across the full range of pressures. Results are plotted in Fig. 4(a), showing that 24 and 30-Å separations (blue and green data points) have identical freezing temperatures at all pressures as the "unconfined" configuration (black data points). This allows us to conclude that the freezing temperature enhancement reported previously, e.g. for 24 and 30-Å capillary bridges in Fig. 3, are due to pressure alone. The 18-Å setup (red data points in Fig. 4(a)) exhibits a significant increase in freezing temperature as a result of the confined geometry. This behavior is indeed evident in the departure of the 18-Å capillary bridge freezing data point from the linear trend in Fig. 3. This is why we have excluded the 18-Å capillary bridge data from the slope analysis in the previous section.

### 3.4 Freezing locations relative to the air-water interface

We now consider whether there is an analogous confinement effect along the $x$-axis, between the two air-water interfaces of our capillary bridge simulations. A narrow capillary bridge configuration can be seen in Fig. 1(d). These narrow capillary bridges are 30 Å wide, while the capillary bridges used in our previous results (henceforth referred to as "wide capillary bridges") are 60 Å wide. The total surface area of substrate–water contact is kept constant between the narrow and wide capillary bridges by

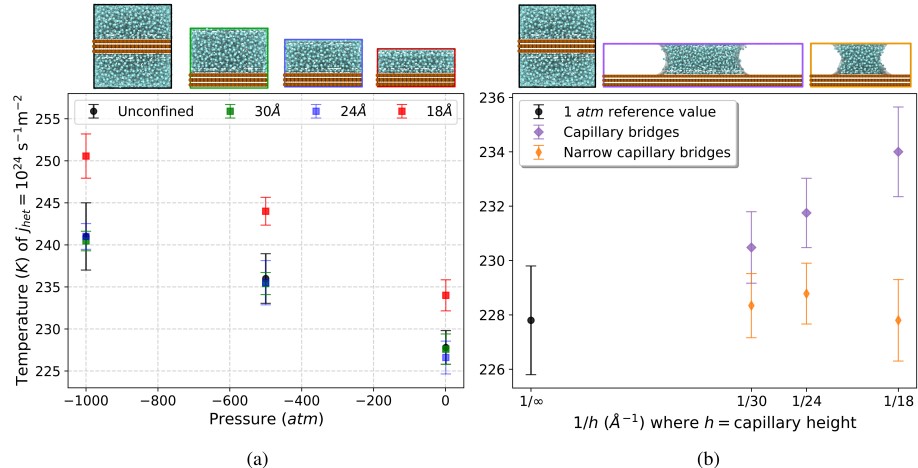

**Figure 4.** (a) Heterogeneous freezing temperature versus pressure with varying separation between substrates, illustrating the effect of confinement along the $z$-axis. We see that confinement influences the ice nucleation rate for only the 18-Å configuration (red squares). The 24-Å (blue squares) and 30-Å (green squares) confinement configurations exhibit the same freezing temperatures at all pressures as the "unconfined" reference simulations (black circles). (b) Heterogeneous freezing temperature versus inverse capillary-bridge height for varying widths of the capillary bridge, showing the effect of confinement along the $x$-axis between air-water interfaces. Confining the water within a narrow capillary bridge (orange diamonds) suppresses the increase in freezing temperature that is seen in the capillary bridges that are twice as wide (purple diamonds).

doubling the $y$-dimension of the narrow capillary simulation box. Figure 4(b) shows heterogeneous freezing rate temperature as a function of $1/h$ for the wide capillary bridges (purple diamonds), compared with narrow capillary bridges (orange diamonds). We observe that confining the water within a narrow capillary bridge eliminates the temperature enhancement from Laplace pressure that is observed in the wide capillaries. This result may be connected to the apparent suppression of ice nucleation near the air-water interface, as discussed below.

Figure 5(a) shows the $x, z$-plane locations of ice nucleation events within the wide 30-Å tall capillary bridge viewed from the side. The air-water interface is shown in red shading. Figure 5(b) shows the combined locations of ice nucleation events in the wide 24-Å and 30-Å tall capillary bridges, as viewed from above in the $x, y$-plane. In this case, the red shading indicates the position of the air-water interface at the substrate. We see that nucleation is preferred near the substrate as expected, but also that it is suppressed near the air-water interfaces. In Fig. 5(c) we plot freezing probability density as a function of distance from the nearest substrate surface using all freezing events in the wide 24-Å and-30 Å tall capillary bridges. Because this distribution is asymmetric (being affected by the substrate on one side) we fit a gamma distribution to the data to find that 99% of nucleation events occur at distances between 3.2 and 6.9 Å from the substrate, along the $z$-axis. Figure 5(d) shows the probability density of ice nucleation as a function of distance from the nearest air-water interface using all freezing events in the wide 24-Å and 30-Å capillary bridges. Analyzing this spatial distribution of ice nucleation events, we observe that ice nucleation never occurs within approximately 10 Å of the air-water interface. This lack of heterogeneous ice nucleation in the

immediate vicinity of the air–water interface could be related to premelting at the ice–vapor interface, as described by Qiu and Molinero (2018).

The narrow capillary bridges being 30-Å wide implies that most of the water molecules are positioned within 10 Å of an air-water interface where no ice nucleation events are observed. In all of the narrow capillary bridges (orange diamonds in 4(b)) the heterogeneous freezing temperature is the same as the 1 atm reference temperature (black data point in Fig. 4(b)). Furthermore, the enhancement in freezing temperature previously seen in the wide 18-Å tall capillary due to confinement between the substrate layers is eliminated inside the narrow 18-Å tall capillary bridge.

Suppression of ice nucleation in the vicinity of flat air-water interfaces has been noted by Haji-Akbari et al. (2014) using the mW model. They explain this effect through the observation that ice embryos near the interface tend to be less spherical compared to in the bulk, thus imposing a larger ice–liquid surface energy term that inhibits nucleation. For the more detailed TIP4P/Ice model, Haji-Akbari and Debenedetti (2017) saw that ice nucleation was not only suppressed within 10 Å of the air-water interface but also showed evidence of higher ice nucleation rates in the sub-surface region of the interface compared to in the bulk. In Figure 5(d), we see that the freezing locations in the wide capillary bridge simulations hint at a slight preference for ice nucleation to occur between ∼20 and 25 Å from the air-water interface. However, when we repeat these simulations with an even wider (120-Å) capillary bridge we find that this propensity for nucleation in the sub-surface region near the air–water interface is not broadly observed for all geometries (see Appendix B). Whether or not the ML-mW exhibits any surface freezing propensity is inconclusive from our results.

In summary, confinement between the ice-nucleating substrates at scales smaller than 20 Å tends to enhance nucleation in the ML-mW model, whereas confinement between air-water interfaces on scales smaller than 30 Å tends to inhibit heterogeneous nucleation.

## 4   Discussion

From what we understand about ice nucleation, a supercooled water droplet in a cloud at a given temperature can be expected to freeze within a time frame dictated by the heterogeneous ice nucleation rate of the particle surfaces it may be in contact with. However, the factors that influence the heterogeneous nucleation rate are not all understood. Research on heterogeneous freezing in the atmosphere is performed on scales ranging from full clouds, using in-situ and remote sensing measurements to characterize aerosols and to identify the presence of ice in clouds; to the scale of single water droplets in laboratory studies and computer simulations to investigate the mechanisms that lead to freezing. Most measurements and computational results are interpreted in the context of classical nucleation theory, allowing the roles of temperature and time in the nucleation process to be understood (e.g., Niedermeier et al., 2011). Most often, it is assumed that singular properties dominate over the stochastic time dependence, and it is therefore typical in cloud physics to characterize ice nucleating particles in terms of their freezing temperature (e.g., Hoose and Möhler, 2012; Frostenberg et al., 2022). Knowledge of the freezing efficiency of ice nucleating particles is then used to understand the formation of ice in clouds (e.g., Yang et al., 2013; Fu and Xue, 2017). Their

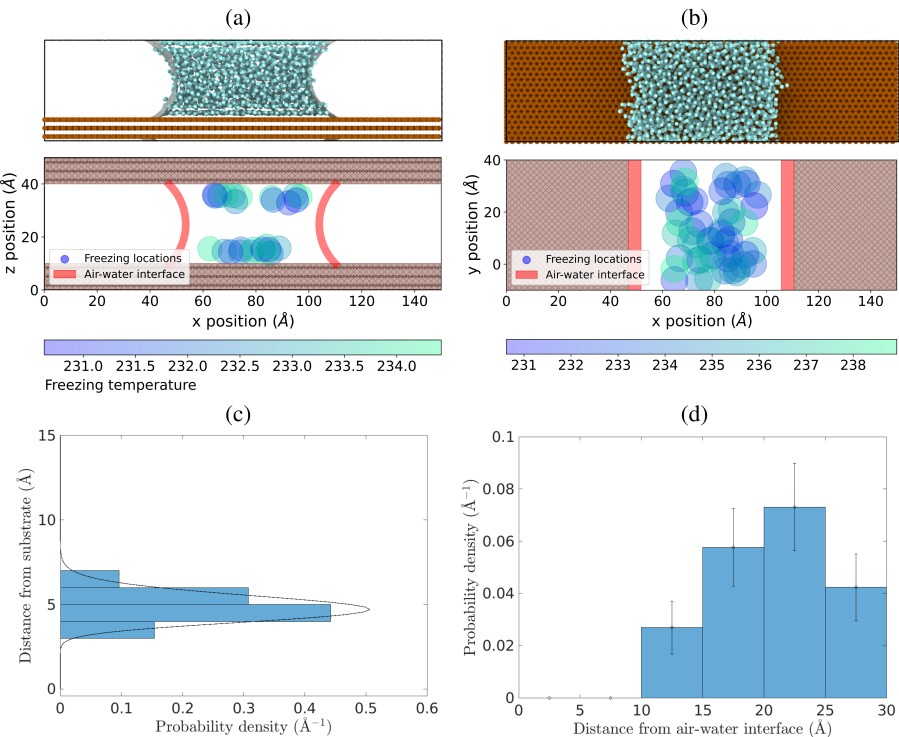

**Figure 5.** Spatial locations of ice nucleation within water capillary bridges. (a) Locations in a 30 Å tall capillary bridge, in the x-z plane. (b) Locations of freezing events in the x-y plane for both the 24 Å and 30 Å capillary bridges combined, viewed from above. (c) Probability density of ice nucleation initiating a distance away from the substrate, using freezing locations inside the 24 Å and 30 Å capillary bridges. The dashed line is a gamma distribution fit. (d) Probability density of ice nucleation initiating a distance away from the air-water interface (red shading in Fig. (a) and (b)) freezing locations inside the 24 Å and 30 Å capillary bridges.

representation in coarse-resolution models even has implications for prediction of Arctic amplification in the climate problem (Tan et al., 2022).

This study has focused on how, for a fixed nucleation rate, the heterogeneous freezing temperature increases with decreasing pressure. Our key result is that the temperature corresponding to a certain ice nucleation rate increases linearly with the magnitude of negative pressure. Specifically, regardless of how negative pressure is created in the system, e.g. barostat or Laplace pressure, the change is described by the approximation $\Delta T / \Delta P = T_m \Delta v_{ls} / l_f$ (see Eq. (1)). This is certainly not true across a broad range of pressures (e.g., Bianco et al., 2021; Espinosa et al., 2016); however, for atmospheric processes

it is likely that only the negative pressure range of 1 atm to $-1000$ atm, which was investigated here, is relevant. Therefore, the linear approximation can serve as the basis for a straightforward parameterization of the pressure effect. Essentially, the temperature increase for heterogeneous freezing is determined in large part by the volume difference between liquid and ice.

For real water, using the values listed in Table 1, we can estimate that the slope $(\Delta T / \Delta P)$ is $7.3 \times 10^{-8}$ K Pa$^{-1}$ = $7.3 \times 10^{-3}$ K atm$^{-1}$.

To understand the implications this pressure effect could have in atmospheric ice nucleation, consider a hypothetical case where atmospheric cloud droplets containing a certain type of ice-nucleating particle (IN) are likely to freeze within a second at $-20$ °C. This would correspond to a nucleation rate of what we shall call $J_{atmospheric}$. The parameterization given by Eq. (1) states that if the water in contact with the ice nucleating substrate is under a tension of $-500$ atm, then $\Delta T_{het} \approx 3.7$ K. Thus, the same nucleation rate $J_{atmospheric}$ at which freezing occurs within a second, would instead be encountered at $-16.3$

410 °C. A common parameterization for the concentration of ice forming nuclei is $n_{IN} \propto \exp(\beta \delta T)$, where $\delta T = T_m - T$ and $\beta \approx 0.6$ K$^{-1}$ (Pruppacher and Klett, 1997, Sec. 9.2). It follows that the enhancement in the concentration of ice nucleating particles when pressure is reduced by $\Delta P$ is $n_{IN,\Delta P}/n_{IN} = \exp(\beta \Delta T)$, where $\Delta T$ is obtained from $\Delta P$ using Eq. (1). An order-of-magnitude increase in concentration results from $\Delta T = 3.8$ K, very close to the value suggested above for $-500$ atm. More recent measurements have led to refinements in parameterizations of $n_{IN}$, but the general exponential trend with $\delta T$ still

holds in many contexts (DeMott et al., 2010). Thus, these moderate differences in the temperature at which IN are active are certainly worth attention in the context of mixed-phase clouds.

     Natural examples abound with water under tension, having negative pressures in the range of 100s of atmospheres, within the range explored in this paper. For example, changes in pressure within mineral inclusions have been shown to lead to significant changes in the conditions for ice-water coexistence, and even to the superheating of ice (Roedder, 1967). Soils consist of a

420 hierarchy of particle sizes that are bound through capillary tension, with similar pressure ranges being present (Seiphoori et al., 2020). Negative pressures in trees, and even synthetic trees, reach negative pressures of 100s of atmospheres (Wheeler and Stroock, 2008). Negative pressures can also be generated through dynamic means. For example, droplets impacting solid or liquid surfaces can experience significant pressure perturbations (Cheng et al., 2022). In fact, mechanical impact has been implemented as a method to initiate ice nucleation to avoid the persistence of supercooled liquid in phase-change thermal

storage systems (Wang et al., 2022). Conversely, imposing isochoric conditions has been shown to greatly increase the stability of supercooled water so that it can be used for cryopreservation (Powell-Palm et al., 2020). One can speculate that there could be connections to contact nucleation or even formation of ice from collisions between supercooled droplets (Alkezweeny, 1969) or breakup of supercooled raindrops that contain ice nucleating particles that otherwise would not be active, save for large, transient negative pressures (James et al., 2021). It has been observed repeatedly that ice generation in clouds is correlated

with the presence of large drops (Rangno and Hobbs, 1991; Lance et al., 2011), and it is worth noting that such drops are exactly what is needed to allow significant collisional growth or drop breakup. These will be exciting ideas to explore in future research.

     Other perspectives on ice nucleation can also be related to the pressure and liquid–ice molar volume difference results presented here. Baker and Baker (2004) argued that freezing in pure water occurs at the temperature at which its compressibility,

with associated density fluctuations, reaches a maximum. The results were for atmospheric pressure, but the perspective that local densities of water and ice drive ice nucleation, rather than specific interfacial properties, is consistent. Insofar as pressure and water activity both contribute to the chemical potential difference between liquid water and ice (Němec, 2013), our results

are consistent with previous efforts connecting heterogeneous nucleation to water activity (Koop et al., 2000; Knopf and Alpert, 2013). The effect of pressure on nucleation rate can be interpreted in terms of water activity; for example, (Knopf and Alpert, 2013) have shown via extensive laboratory experiments with a range of materials, that nucleation rates scale with water activity. This further extends the findings of Koop et al. (2000) that freezing and melting are related to pressure, following the water activity. Freezing temperature depression due to positive pressure in water was measured experimentally by Kanno et al. (1975) and decrease in ice nucleation rate due to positive Laplace pressure in nano-scale water droplets was demonstrated by Li et al. (2013), both leading to the intuitive notion that negative Laplace pressure will cause an increase in freezing temperature.

Creating Laplace pressures large enough in magnitude to enhance ice nucleation temperatures involves geometric configurations on nanometer spatial scales, which also may introduce confinement effects into the system. Thus, one must consider whether or not enhancement in ice nucleation due to Laplace pressure will be suppressed or further enhanced by the specific geometry at play. For example, water in the xylem of vascular plants should be under significant negative pressure, yet Lintunen et al. (2013) shows a tendency for ice nucleation to be suppressed. Our simulations in Sections 3.3 and 3.4 show that confinement between flat substrate surfaces can lead to enhanced nucleation rates, whereas confinement between air–water interfaces can suppress ice nucleation. Furthermore, other examples of confinement exist, such as experimental measurements compiled in Marcolli (2014) which show a depression in freezing temperatures of water inside completely filled pores with nanometer scale diameters. The nature of the confinement present is likely an important consideration in interpreting various findings. Water confined between flat surfaces separated by less than ∼2 nm may experience ice nucleation enhancement due to the density oscillations induced by the flat interfaces; however, the curved geometry in a cylindrical pore of the same size would not produce this density oscillation effect (Cox et al., 2015; Bi et al., 2016; Lupi et al., 2014).

Even though the findings presented in this study describe the effect of negative pressure on heterogeneous ice nucleation rates, there are still open questions remaining that involve the thermodynamic properties of water at interfaces, under confinement, and across greater regions of waters phase diagram. Key questions of interest to refine and expand upon the results of this study include; (1) simulating different substrates to observe changes, if any, in the slope of heterogeneous nucleation rate coefficient lines, (2) exploring liquid water and ice properties near substrates to determine whether they alter the parameters in Eq. (1), (3) assessing the level of surface freezing propensity present in the ML-mW water model and how this factors into the confinement effects seen in this study, and (4) investigating the interplay between cavitation and ice nucleation in this doubly metastable regime of water.

# 5 Concluding Remarks

Using molecular dynamics simulations of heterogeneous ice nucleation, we demonstrate that negative pressure within supercooled water allows for a given ice nucleation rate to occur at higher temperatures. The increase in heterogeneous freezing temperature with negative pressure can be estimated as linear in nature, which lends support to the use of a linear approximation that depends on the latent heat release and the molar volume difference between liquid and ice to predict the slope. This

approximation, given by Eq. (1), works particularly well for homogeneous nucleation rates; it is acceptable for heterogeneous nucleation, but may need adjustment to provide better quantitative agreement.

    To observe this pressure-dependent trend in heterogeneous nucleation rate, we first use a simulation setup containing an ice-nucleating substrate immersed in water. A barostat is applied to this system to probe pressures of 1 atm, $-500$ atm, and $-1000$ atm. Next, we create a capillary bridge configuration where negative pressure is introduced into the water without applying a

barostat, but instead through negative Laplace pressure inside a water capillary bridge formed between the substrate surfaces. The magnitude of negative Laplace pressure within the capillary bridge depends on the curvature of the water surface, set by the height of the capillary bridge. A range of negative Laplace pressures is sampled by using heights of 18, 24, and 30 Å. The observed enhancement in heterogeneous freezing temperature within the capillary bridges is consistent with Eq. (1).

    These simulations demonstrate that nano-scale water surface curvature, corresponding to negative Laplace pressures of

hundreds of atmospheres, can result in several-Kelvin increases in heterogeneous freezing temperature. In mixed-phase clouds, such changes in freezing temperature would lead to a considerable increase in active ice-nucleating particle concentrations. Furthermore, our findings indicate that any process which produces substantial negative pressure perturbations in supercooled water can increase the rate of ice formation. Thus, dynamic processes such as droplet collision or breakup may warrant further investigation as potential ice-nucleation mechanisms.

**Appendix A: Molecular Interaction Potential**

    Interactions between water molecules and other water molecules, as well as interactions between water molecules and substrate molecules are all described using versions of the Stillinger-Weber interaction potential (Stillinger and Weber, 1985). When used for interactions involving water molecules, this potential is coarse-grained, meaning that the oxygen and hydrogen atoms are combined into one atom. The bonds between water molecules are then represented using a three-body potential, $\phi_3$, which is

a function of the angle $\theta_{ijk}$ formed between every set of three water molecules ($i$, $j$, and $k$). This three-body potential creates a preference towards water molecules adopting a bond angle of approximately 105 degrees, set by the $\cos\theta_0$ parameter in Table A1. A two-body interaction term, $\phi_2$, applies forces that are dependent on the radial distance between two atoms $r_{ij}$.

$$\phi_2(r_{ij}) = A\epsilon \left[ B(\frac{\sigma}{r_{ij}})^p - (\frac{\sigma}{r_{ij}})^q \right] \exp\left( \frac{\sigma}{r_{ij} - a\sigma} \right) \tag{A1}$$

$$\phi_3(\theta_{ijk}, r_{ij}, r_{ik}) = \lambda\epsilon[\cos\theta_{ijk} - \cos\theta_0]^2 \exp\left( \frac{\gamma\sigma}{r_{ij} - a\sigma} \right) \exp\left( \frac{\gamma\sigma}{r_{ik} - a\sigma} \right) \tag{A2}$$

The parameters used for the interaction between water molecules and the substrate are summarized in Table A1. The cutoff distance where forces between molecules goes to zero is $a\sigma$. Note also that interactions between water molecules and substrate molecules do not have a three-body contribution, and are only influenced by the two-body potential term.

    In creating a suitable interaction potential for an interaction between ML-mW water and the substrate, we start with the ML-mW–ML-mW parameters and, as in the methods of Fitzner et al. (2015), only the $\epsilon$ and $\sigma$ values are adjusted to produce

our ML-mW–Substrate interaction. Given that the value of $\epsilon$ we have used is larger than that of the mW–Carbon interaction,

**Table A1.** Parameters of the interaction potential between water and the substrate, and for water-water interactions, for the mW model and the ML-mW model. The Stillinger–Weber interaction potential is given by Eq. (A1).

| | mW–Carbon (Lupi et al., 2014) | ML-mW–Substrate | mW–mW (Molinero and Moore, 2009) | ML-mW–ML-mW (Chan et al., 2019) |
|---|---|---|---|---|
| $\epsilon$ (Kcal/mole) | 0.13 | 0.35 | 6.189 | 6.855473 |
| $\sigma$ (Å) | 3.2 | 2.2 | 2.3925 | 1.884015 |
| $a$ | 1.80 | 2.124872 | 1.80 | 2.124872 |
| $\lambda$ | 0.0 | 0.0 | 23.15 | 24.673877 |
| $\gamma$ | 0.0 | 0.0 | 1.20 | 1.207943 |
| $\cos\theta_0$ | 0.0 | 0.0 | −0.33 | −0.279667 |
| $A$ | 7.049556277 | 7.111598 | 7.049556277 | 7.111598 |
| $B$ | 0.6022245584 | 1.991526 | 0.6022245584 | 1.991526 |
| $p$ | 4.0 | 4.011214 | 4.0 | 4.011214 |
| $q$ | 0.0 | 0.0 | 0.0 | 0.0 |
| $tol$ | 0.0 | 0.0 | 0.0 | 0.0 |

the smaller contact angle that we see for ML-mW–Substrate is consistent with Bi et al. (2016) and Cox et al. (2015), where higher values of $\epsilon$ were used to increase hydrophilicity of the Carbon-mW interaction potential Bi et al. (2016).

## Appendix B: Freezing locations in an extra wide capillary

Prompted by a reviewer comment, we constructed a 120-Å wide capillary bridge to look at where in relation to the air-water interfaces ice nucleation events tend to occur. Using 50 ice nucleation events, we see that they again avoid the immediate ∼ 10 Å vicinity of the air–water interface. However, the slight preference for freezing in the region just beyond 10 Å from the air–water interface that was noticed in the 60-Å wide capillary bridges is no longer pronounced in this 120-Å wide capillary bridge. Oscillations in the water density field may play a role here (Cox et al., 2015; Bi et al., 2016; Lupi et al., 2014), making this an intriguing subject for further investigation.

*Code and data availability.* LAMMPS is free and open-source software developed by Sandia National Laboratory and a large user community. Scripts to reproduce simulations in LAMMPS and data going into Figures can be found at http://doi.org/10.37099/mtu.dc.all-datasets/41

*Author contributions.* ER, WC, and RAS designed the study; ER, TL, and IN set up the simulations; ER ran the simulations; All authors contributed to interpreting the results and writing the paper.

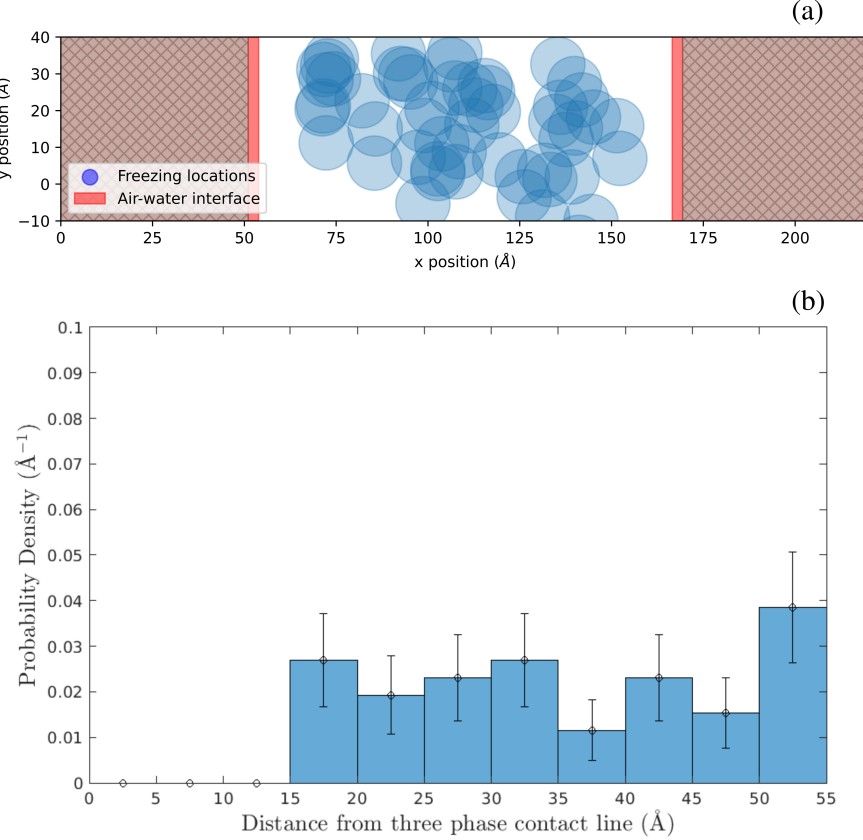

**Figure B1.** (a) Ice nucleation locations in the x–y plane of a 120-Å wide capillary bridge. (b) Probability distribution of freezing events as a function of distance from the nearest air-water interface in a 120 Å wide capillary bridge. The figure contains 50 data points.

*Competing interests.* No competing interests are present

*Acknowledgements.* Funding from NSF grant AGS-2019649 is gratefully acknowledged. TL acknowledges support from NSF through award CBET-2053330. IN acknowledges NSF (DMR-1944211). The High-Performance Computing Shared Facility (Portage) at Michigan Technological University was used in obtaining results presented in this publication. We acknowledge the LAMMPS community for helpful discussions and support.

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
