# Peer review of "Molecular simulations reveal that heterogeneous ice nucleation occurs at higher temperatures in water under capillary tension"

_EGUsphere, 2023_

## Author Comment (AC1)

**Author's Response**

**General Response**

Thank you for the opportunity to submit a revised manuscript with the title "Molecular simulations reveal that heterogeneous ice nucleation occurs at higher temperatures in water under capillary tension" to Atmospheric Chemistry and Physics. We appreciate the time and effort that the editor and the two reviewers have dedicated to providing valuable feedback on our manuscript. In particular, we are gratified that the reviewers find that "These are interesting results that are worth publishing in ACP" (Reviewer 1) and "The topic addressed in this paper is of significant importance" (Reviewer 2). We have addressed the issues raised by the reviewers without exception. Our sense is that their comments have resulted in significant improvements to the manuscript. We hope that the current manuscript will be considered acceptable as a step in bringing attention to this interesting and important problem.

Below is a general response to the reviewers' key concerns, followed by point-by-point responses to each reviewer's comments:

Our study explores the hypothesis that negative pressure in supercooled water can result in heterogeneous ice nucleation at higher temperatures than expected at ambient pressure. A linear approximation, Eq. (1) of the manuscript, was previously shown to describe the slope of homogeneous ice nucleation rate in P–T coordinates. We now aim to assess whether the same linear expression can be applied to heterogeneous freezing rates. Finally, we want to know if negative pressure produced by Laplace pressure from a curved air-water interface can drive heterogeneous nucleation to occur at higher temperatures in accordance with Eq. (1).

We will first summarize the key revisions that have been made to the manuscript In response to both reviewers' feedback.

    1.  In the revised manuscript, we give specific context for what types of freezing temperature enhancements due to negative pressure would be considered significant to

atmospheric heterogeneous ice nucleation. We explain that a moderate increase in the nucleation rate associated with nano-scale curvature, corresponding to a ~4 K increase in the freezing temperature, can result in an order of magnitude increase in ice-nucleating activity in the atmosphere. This added context gives the reader greater appreciation for the significance of these seemingly moderate enhancements in freezing temperature due to negative pressure. **(See 3rd paragraph of Introduction, and 3rd paragraph of Discussion)**.

2. We have revised the text in several places to clarify that the $(dT/dP)_{freeze}$ slope is not purported to be linear, but it can be approximated as linear in the context of atmospherically relevant conditions. The revised manuscript also thoroughly describes, with support from literature, that approximating $(dT/dP)_{freeze}$ as parallel to $(dT/dP)_{melt}$ is valid at negative pressures, even if less so for positive pressures. These two arguments together provide justification that the linear approximation given by Eq. (1) is appropriate to use in the negative pressure regime, within the scope of freezing enhancement due to negative pressure. **(See the Introduction section following Eq. (1)).**

3. We refine our discussion around the observed steepness of the ML-mW heterogeneous freezing line. Our assessment is that the heterogeneous freezing line has a similar $(dP/dT)_{freeze}$ trend as homogeneous nucleation and may lend itself well to a linear parameterization. However, the slope of the ML-mW model heterogeneous freezing line is steeper than that of the homogeneous line, indicating that other factors in addition to the variables used in Eq. (1) may have a role in shaping the slope of this line. Incorporating thoughtful insights from the reviewers, we have extended our discussion of the possible factors at play here, and a deeper investigation of this topic is recommended for future work. **(See paragraph 2, 5 & 6 of Discussion)**.

4. Follow-up simulations in response to a reviewer comment have prompted us to revise our interpretation regarding the trend in ice nucleation to prefer the sub-surface region near the air-water interface. We doubled the width of the capillary bridge and simulated 50 ice nucleation events to see if the findings were reproduced. We found that ice nucleation still avoided the air-water interface, but the new simulations did exhibit the same propensity for freezing in the sub-surface region. We have revised our conclusions to reflect this new information. **(See Paragraph 4 of Section 3.4, and Appendix B)**.

5. Keeping our specific research objectives in mind, we are also aware that the findings presented here lend themselves to many fascinating and connected topics involving the thermodynamic properties of water at interfaces, under confinement, and across greater regions of the phase diagram, all of which are flush with many exciting open questions. To address these important connections while still maintaining a focused scope for this manuscript, we have created an extended conclusions/discussion section in the revised manuscript to highlight key open questions that require further investigation to refine and expand upon the results of this study. Those topics include **(See last paragraph of Discussion)**:

- Simulating different substrates to observe changes, if any, in the slope of heterogeneous nucleation rate coefficient lines,
- Exploring liquid water and ice properties near substrates to determine whether they alter the parameters in Eq. (1),
- Assessing the level of surface freezing propensity present in the ML-mW water model and how this factors into the confinement effects seen in this study, and
- Investigating the interplay between cavitation and ice nucleation in this doubly metastable regime of water.

Here we provide a summary of how we have addressed each reviewer's key concerns in the responses to follow, with detailed elaboration provided in the point-by-point responses:

Reviewer #1: We understand that the primary concern is applying the ML-mW water model in negative pressure regimes (where there is no experimental data) without first validating the model against experimental data in positive pressure regimes. We have addressed this concern by (1) referencing simulation studies in the positive-pressure regime, which show that the homogeneous freezing curves of mW water model, as well as other more detailed water models, exhibit the same qualitative behavior at positive pressures as observed by the experiments of Kanno et al. (1975). (2) We performed an additional simulation of ML-mW homogeneous freezing at +500 atm to confirm that the model is in both qualitative and quantitative agreement with Kanno's experimental results. (3) We also point out that the equilibrium melting point line produced by the ML-mW model matches the experimental line at negative pressures, which provides further confidence in the performance at negative pressures. Points (1) and (2) have been included in the revised manuscript **(See first paragraph of Methods)**.

Secondly, Reviewer #1 notes that the approximation of Eq. (1) does not match experimental data in the positive pressure regime by Kanno et al. (1975). We mainly address this concern by providing a much more thorough justification that this approximation is valid only in pressure regimes where the freezing line and melting line are parallel, which emerging literature has shown is true for pressures less than 500 atm. In this light, we point out that the approximation of Eq. (1) shows excellent agreement with Kanno et al. (1975) if one restricts their analysis to pressures below magnitude of 500 atm. For example, at 200 atm, Eq. (1) predicts a freezing point depression of 1.5 K, which is exactly reflected in the Kanno et al. homogeneous freezing curve. **(See sixth paragraph of Introduction)**.

Reviewer #2: We understand this reviewer's primary concerns are lack of clarity in the methods, overlooking important studies in the literature that are highly relevant to this work, and that the scope of our results should be clearly justified and communicated. Additionally, this reviewer points out some very insightful physical considerations that we have incorporated into the interpretation of our results throughout the manuscript. To address these remarks, we have responded to all the reviewer's comments on the methodology and updated the Methods section in the manuscript. We have studied the provided references and included them in our

interpretation. We have added a section in the Introduction to clarify the scope in which our results can be applied (namely, in the negative pressure regime of 1 atm to –1000 atm) **(See sixth paragraph of Introduction)**. For comments where we are not able to provide conclusive answers from our available data, we have included these topics in our discussion of important future work **(See last paragraph of Discussion)**.

**Additional simulations provided to address reviewer concerns:**
(All new data shall be added to the database repository associated with this manuscript)

1 - ML-mW homogeneous freezing at +500 atm (30 runs)
  ● Increase confidence that ML-mW follows water's homogeneous freezing behavior in positive-pressure regimes where experiments exist, thereby increasing confidence in results from negative-pressure regimes.
2 - Double width of capillary bridge (50 runs)
  ● Determine that with a greater distance between air-water surfaces, nucleation does not show a preference for regions near the air-water interface.
3 - Double z-axis of heterogeneous freezing (20 runs)
  ● Confirms that z-axis confinement is not impacting ice nucleation in our "unconfined" simulation cells.

**Reviewer #1**

This study investigates the heterogeneous freezing temperature increase at negative pressure with the mW and MLmW water models. Simulations have been carried out at negative pressures of -500 and -1000 atm with water that was in contact with a hydrophilic substrate that promotes ice nucleation. These simulations showed an approximately linear increase in heterogeneous ice nucleation temperature with decreasing pressure. Moreover, water freezing was simulated in water capillary bridges of heights from 3.0 to 1.8 nm. Here, an approximately linear relationship between the capillary bridge width and the heterogeneous ice nucleation temperature was found for unconfined water and water capillary bridges of 3 and 2.4 nm. For capillary bridges of 1.8 nm, an even increased nucleation rate was simulated. Based on these results, a linear relationship between pressure and heterogeneous freezing temperature was derived. This linear relationship was proposed to serve as a basis to estimate the pressure effect on heterogeneous freezing. Moreover, the simulations were used to investigate the location of ice nucleation. It was found that heterogeneous ice nucleation does not occur in the regions within 1.0 nm of the air-water interface.

These are interesting results that are worth publishing in ACP. However, there are weaknesses in the discussion of the results. The ability of the mW and the MLmW water models to describe the pressure dependence of homogeneous and heterogeneous ice nucleation has not been assessed properly. Nevertheless, a parameterization derived from the simulation results was proposed to predict the pressure dependence of freezing temperatures at negative pressure. However, such a recommendation is only justified when the MLmW model is able to describe the pressure dependence of ice nucleation correctly. The comparison to experimental data (Kanno, 1975) reveals that the proposed pressure dependence underpredicts the freezing temperature depression at positive pressure (see specific comments). It should be explained why the proposed pressure dependence should be accurate at negative pressure when the model is not able to describe the pressure dependence at positive pressure correctly. Similarly, the increased nucleation rate in the water capillary bridge is not critically reviewed in view of experiments that show the opposite trend (see e.g. Marcolli, 2014, for a compilation of experiments).

**References**

Kanno, H., Speedy, R. J., and Angell, C. A.: Supercooling of water to -92°C under pressure, Science, 189, 880–881, https://doi.org/10.1126/science.189.4206.880, 1975.

Marcolli, C.: Deposition nucleation viewed as homogeneous or immersion freezing in pores and cavities, Atmos. Chem. Phys., 14, 2071–2104, https://doi.org/10.5194/acp-14-2071-2014, 2014.

**Technical Corrections**

Check mark indicates that the technical correction was implemented in the revised manuscript.

- ☑ Table captions should be above the tables.
- ☑ References: Journal titles are not abbreviated according to the journal's guidelines. Also, they are not consistently formatted: Some are with and some without DOIs; some use capital letters in article titles while others do not.
- ☑ Line 6: There are two "from". One should be deleted.
- ☑ Line 116: "feasibly be achieved": either just "feasible" or just "be achieved".
- ☑ Line 125: the abbreviation"NVT" should be explained.
- ☑ Line 139– 41: this sentence should be improved.
- ☑ Line 179: "comparison with Rosky et al." instead of "comparison from Rosky et al."
- ☑ Line 196:  Do you mean Equation (2) instead of Equation (1)?  Moreover, in most parameterizations, the pre-factor "A" is different for homogeneous and heterogeneous ice nucleation.
    - **Response:** We are referring to Eq (1) in these cases because we are assessing how well the linear approximation given by Eq (1) agrees quantitatively with the various freezing lines. With regard to the pre-factor "A", we agree that is an important distinction to make and we have added the subscript "$A_{het}$" when referring to heterogeneous nucleation rate.
- ☑ Lines 210–211: this sentence is incomplete.
- ☑ Line 276: "a capillary bridge" or "capillary bridges".
- ☑ Figure 4: Panel b of this figure is explained only after Fig. 5. The manuscript should be reorganized in a way that the figures are explained in the right sequence.
- ☑ Figure 5: The legend and axis numbers and the colour scale numbers are two small and should be increased.
- ☑ Line 389: "adobpting": remove the "b"
- ☑ Line 460: the paper title is not correctly displayed.

**Comments**

- ☑ Lines 14–16: "and shows a preference for nucleation in the region just beyond 10 Å": do you refer here to the distance from the air-water interface or the distance from the substrate? This should be clarified.

**Incorporated into Revision:** Yes

We are referring to the region just beyond 10 A from the air-water interface. However, upon conducting followup simulations in response to a comment from Reviewer #2, we have removed this conclusion from our interpretation. (See Appendix B of revised manuscript).

☑ Lines 52–55: Here, it is stated that the slope of the freezing temperature as a function of pressure is parallel to the slope of the melting line. However, inspection of Fig. 1 shows that this is not the case. A parallel relationship would only be fulfilled if the enthalpy of fusion and the molar volume difference were independent of temperature.

**Incorporated into Revision:** Yes. (Paragraph 5 and 6 of Introduction)

We believe the reviewer is referring to Figure 2, and we agree with their statement that approximating a parallel relationship between the melting and freezing lines requires certain conditions to be met.

The derivation of our approximate form of $(dT/dP)_{freeze}$ includes the following assumptions: the molar volume of water and ice is constant with pressure; the temperature dependence of latent heat can be neglected; and surface tension, density, and kinetic prefactor remain constant for small changes in temperature and pressure. For atmospheric applications, the temperature and pressure changes encountered are small enough to justify these approximations. These details of the derivation can be found in Appendix C of Rosky et. al. (2022), but we have also added a sentence to our revised manuscript explicitly stating which variables have been approximated as constant.

More importantly, Rosky et al. (2022) and other studies suggest that the homogeneous freezing line and the melting line become parallel in the negative pressure regime. For example, in the revised manuscript we add references to a recent simulation study by Montero de Hijes (2023), from which we quote:

"...our results suggest that the homogeneous nucleation line should be parallel to the coexistence line when pressure is below ~500 bars (while at higher pressure they are not)." (Montero de Hijes, 2023)

To emphasize that certain assumptions have been made to suit the pressure regime we study, we have added significant elaboration to the discussion introducing Eq. (1) of our manuscript.

Regarding the trends in slope shown in Fig. 2: We can say with confidence that, in this pressure regime, the homogeneous freezing lines (blue) are nearly parallel to the melting line for both water models. However, the heterogeneous freezing line (red) shows some additional steepness in the ML-mW which is discussed in the manuscript.

Montero de Hijes, P., R Espinosa, J., Vega, C., and Dellago, C.: Minimum in the pressure dependence of the interfacial free energy between ice Ih and water, The Journal of Chemical Physics, 158, 124503, 2023

☑ Figure 1: in Panels c and d, the substrate is only shown below the water bridge. Is this for clarity or is there no substrate above the water bridge? This should be clarified in the figure caption.

**Incorporated into Revision:** Yes

Due to the periodic boundary conditions, the bottom surface of the substrate is in contact with the water molecules at the top of the simulation cell, thus forming the water bridge. We have clarified this in the figure caption.

☑ Lines 172–175: The surface area of the substrate and the rate at which the system is cooled do not influence $J_{het}$ it is formulated as a function of surface area and time. It just influences the time it takes to freeze in the simulation. This needs to be clarified.

**Incorporated into Revision:** Yes.

We understand the way that this sentence is written in the original manuscript could be confusing to readers. The authors mean to explain that the intensive nucleation rate (#/s/area) that we are able to sample within the timeframe of our simulations is dependent on the substrate surface area and the cooling rate. Thus, the cooling rate and substrate area can be thought of as the two levers in our simulation set-up that adjust the range of nucleation rate coefficients available for sampling (and the corresponding temperature range of those nucleation rate coefficients).

Feedback from reviewers makes it apparent that the way we formulated this sentence in the manuscript is confusing to readers. We have revised the wording of this section to be better communicated.

☑ Lines 201–202: The data shows indeed a slightly non-linear trend. This sentence should be formulated more carefully.

**Incorporated into Revision:** Yes

New language: "Most significantly, we observe that the increase in temperature as a function of pressure for $j_{het}$ can be approximated as linear within the sampling uncertainty, indicating that the use of a linear estimate for $(\Delta T/\Delta P)_{het}$ may be applied to heterogeneous ice nucleation."

Though the data does indeed show slight curvature, the ability to approximate the slope of $(dT/dP)_{het}$ as linear is an important result of our simulations because it provides a simple framework that can be used to estimate the magnitude of freezing temperature enhancement one should expect to observe when supercooled water is subjected to negative pressures, requiring only thermodynamic values evaluated at 1 atm.

☑ Line 206–208: the values are still within the uncertainty bounds, but the slope is not strong enough. This weakness in the simulation should be commented.

**Incorporated into Revision:** Yes. (Paragraphs 2, 5 and 6 of Section 3.1) (Final paragraph of Discussion).

We have expanded on our discussion of this weakness in the revised manuscript, elaborating more on the possible physical sources of this result. We state clearly in our discussion that we are not able to conclusively identify the source of the excess steepness in the ML-mW heterogeneous freezing line in this study, and that this topic merits further exploration.

> ☑ Lines 211–212: "While the linear nature of ΔT/ΔP" is apparent in our results". This is an exaggeration. The results are in agreement with a linear relationship given the uncertainty bounds. This sentence needs to be adjusted in this sense.

**Incorporated into Revision:** Yes

New language: "While our results support that the trend in $(\Delta T/\Delta P)$ can be approximated as linear in the negative pressure regime, …"

> ☑ Lines 213–214: Do you refer here to the values given in Table 1 (last line)? If yes, a reference to Table 1 could be given here.

**Incorporated into Revision:** Yes

Reference to Table 1 added here.

> ☑ Lines 219–220: Here, it is hypothesized that the thermodynamic properties of mW water are less influenced near the substrate compared to the MLmW model. Couldn't this be found out by inspecting the simulation?

**Incorporated into Revision:** Yes. (Final paragraph of Discussion).

We did indeed attempt to inspect this in our simulation, but pinning down the thermodynamic properties of only the water that's influenced by the substrate proved to be quite nuanced and challenging. Thus we feel this is a study best left for future work where more careful and rigorous steps can be taken.

> ☑ Lines 221–225: This paragraph is written as if the MLmW model could correctly predict the dependence of freezing temperature as a function of pressure. This assumption needs to be tested by simulating ice nucleation in the positive pressure range and comparing the results to measurements. Such a comparison can be done for homogeneous ice nucleation. See also general comment.

**Incorporated into Revision:** Yes. (First paragraph of Methods).

The authors agree that the reviewers' request to validate the ML-mW model against experimental data in positive pressure regimes is well-advised. Thus we have performed simulations of homogeneous freezing of the ML-mW model at a positive pressure of 500 atm.

Additionally, we provide references where homogeneous freezing of the mW model has been simulated in positive regimes, showing qualitative agreement with the measurement by Kanno et al. (1975).

This Figure 1 from Lu et al. (2016), shows that the mW homogeneous freezing line ($T_x$) is steeper than that of the melting point line ($T_m$) at positive pressure, in qualitative agreement with the experimental results of Kanno et al (1975):

[Figure]

Additionally, other water models such as TIP4P/Ice are also consistent with Kanno et al.'s experimental results across positive pressures, as shown in this figure from Montero de Hijes (2023):

[Figure]

Note that both these figures demonstrate how the melting and homogeneous freezing lines become nearly parallel in the negative pressure regime.

It is reasonable to expect that the ML-mW model will produce similar results as these models in the positive pressure regime; The great quantitative agreement between the ML-mW melting point line and the experimental melting point line plotted in Figure 2 of our manuscript also gives us increased confidence in its ability to correctly predict the shape of the homogeneous freezing point line. Nonetheless, we agree that an explicit validation of the ML-mW model at positive pressures is valuable.

Using 30 simulation runs at +500 atm (using the same methodology to obtain homogeneous nucleation rates in Rosky et al. (2022)), we observe that the ML-mW homogeneous freezing line experiences a 6K depression in freezing temperature at 500 atm pressure. Keeping in mind that there will not be *exact* quantitative agreement between the model and real water, this result does agree remarkably well with the measurement by Kanno et al., which shows a 5-7K depression in homogeneous freezing temperature at this pressure. (The figure from Kanno et al. can be found on page 16 of this response document).

Kanno, H., Speedy, R. J., and Angell, C. A.: Supercooling of water to -92°C under pressure, Science, 189, 880–881, 1975.

Lu, J., Chakravarty, C., and Molinero, V.: Relationship between the line of density anomaly and the lines of melting, crystallization, cavitation, and liquid spinodal in coarse-grained water models, The Journal of Chemical Physics, 144, 2016.

Montero de Hijes, P., R Espinosa, J., Vega, C., and Dellago, C.: Minimum in the pressure dependence of the interfacial free energy between ice Ih and water, The Journal of Chemical Physics, 158, 124503, 2023.

☑ Figure 3: It should be stated whether the dotted line is a fit line or based on the pressure values given in Table 1.

**Incorporated into Revision:** Yes

The line is a fit to the data. We have added a note to the figure caption and referenced the dotted line in the text as well.

☑ Lines 249–253: The calculation of the Laplace pressure by estimating the contact angle from the simulation is indirect. What is relevant for the Laplace pressure is the radius of the meniscus, which could be directly determined from the simulation. Could this still be done to validate the assumed tension within the capillary bridge?

During this study, we made attempts to directly measure the Laplace pressure inside the capillary bridges using three different methods:

1. Using density to pressure conversion.
2. Using the radius of curvature of the meniscus, as suggested here.

3. Using the measured slope of the dT*h and dT/dP to infer the Laplace pressure (as presented in the manuscript).

None of the three methods are particularly precise, and thus they all agree within their large ranges of uncertainty. We felt that the latter method provided the most informative insight into what the estimated pressures in these capillaries are. We do reference the second calculation we made using the radius of curvature in the manuscript stating that "This value of $\theta$ is consistent with estimates we obtain by measuring the radius of curvature of the capillary bridge air-water interfaces using a method similar to \citet{giovambattista2007}"

Those radii of curvature calculations are shown here. We see that the contact angle and the calculated Laplace pressures are consistent with the values we obtain from the calculations in the manuscript.

[Figure]

Capillary curvature

| Capillary height | 30 A | 24 A | 18 A |
|---|---|---|---|
| Radius of Curvature | -21.24 | -26.4 | -17.9 |
| **Laplace Pressure calculation** | **-311 atm** | **-247 atm** | **-370 atm** |

☑ Figure 5: Reading the figure caption and the text, it seems that Panels a and b show the same simulation viewed in different 2D projections. Yet, the colour scale in the panels are different: in Panel a, it is from 231–234 K and in Panel b, it is from 231–238 K. Is this a mistake? Please explain.

**Incorporated into Revision:** Yes

Since the 24 and 30 A tall capillaries have the same width as viewed from above, we are able to show the freezing locations from both capillaries in the same figure in panel b. For panel a, we have only shown the 30 A capillary. Thus the range of freezing temperatures is slightly different between the two. We have added this information in the figure caption and made clear in the text which data is being used to produce each figure.

☑ Figure 5b: the air-water interface (red shaded stripes) seems to be too narrow. It should be broader in the projection because the interface is curved. How was the distance from the air-water interface evaluated? Based on the projection or was the actual distance to the interface taken?

**Incorporated into Revision:** Yes

The red shading in 5(b) is showing the three-phase contact line of the air-water interface with the substrate (not a projection of the full air-water interface). The distance of ice-nucleation from the air-water interface is calculated using this value (distance along x-axis from the three-phase contact line).

The position of the three-phase contact line is calculated using the layer of water molecules within 2 angstroms of the substrate. We divided the y-axis into slices and the positions of the furthest out water molecule along the x-axis in each slice are used to determine the width of the substrate-water interface at that point. Averaging over many timesteps over all the slices gives us the red shading seen in the figure, where the width of the shading shows the standard deviation.

☑ Lines 303–305: A higher nucleation rate for pores narrower than 2 nm is in contradiction with DSC experiments performed on slurries of mesoporous silica materials with pores in this size range, for which no freezing peak at all was observed (see e.g. Marcolli, 2014, for a compilation). This should be commented.

**Incorporated into Revision:** Yes. (Last two paragraphs of Discussion).

This is mainly because the silica nanopores used in experiments are of cylindrical shape, which is different from the confinement geometry in slit pore. The key mechanism for rate enhancement in slit pore is due to density oscillation induced by a flat surface. Curved geometry will destroy this.

We have commented on this in the discussion section. See also our response to the following comment.

☑ Lines 323–325: Here, the simulation results should be critically reviewed in view of the experimental evidence.

**Incorporated into Revision:** Yes. (Paragraph 6 of Section 3.1 and paragraph 6 of Discussion).

We have incorporated this suggestion from the reviewer into the discussion of simulation results in the revised manuscript. We have referenced studies from the reviewers previous comment as well as other experimental results from Evans 1967.

Evans LF. *Two-dimensional nucleation of ice*. Nature. **1967** Jan 28;213(5074):384-5;

☑ Lines 323–327: Here, it is written: "Therefore, the linear approximation can serve as the basis for a straightforward parameterization of the pressure effect." And: "Essentially, the temperature increase for heterogeneous freezing is determined in large part by the volume difference between liquid and ice." These two sentences together imply a linear dependence of the volume difference on pressure. Do the authors really want to imply such a linear pressure dependence? It would be interesting to know whether the simulations support such a linear pressure dependence.

**Incorporated into revision:** Yes

To our understanding, the implication here from these two sentences is that the volume difference between liquid and ice is constant with pressure. The volume difference between liquid and ice determines the slope of dT/dP, which is assumed to be constant. Indeed, in deriving Equation 1 for the slope dT/dP, we explicitly make the assumption that the density of liquid and ice would not change significantly with the pressure/temperature changes dP and dT.

It is known that the density of water does indeed change with temperature and pressure, becoming less dense with decreasing temperature and decreasing pressure and a correction term could be added to the dP/dT approximation to account for this. However, these corrections are very small (indistinguishable within our simulation uncertainty), and can thus be neglected for the purposes of this study.

These topics are again related to the approximations that have been made to suit the moderate negative pressure regime we study. We have addressed this comment in our revision by clarifying that the molar volume difference is assumed constant across the range of temperatures and pressures relevant to this study.

☑ Line 328: If the proposed dependence of freezing temperature on pressure is extrapolated to positive pressure, a freezing point depression of 7.3 K would be expected at 1000 atm. Yet, experimental data by Kanno et al. (1975) show that the freezing point depression is already 7 K at 500 atm and increases to 17 K at 1000 atm. Is there any evidence that the simulations are better in predicting pressure dependence for negative than for positive pressure?

**Incorporated into Revision:** Yes. (Sixth paragraph of Introduction).

In a previous comment, the reviewer asks if the ML-mW water model can be expected to reproduce homogeneous freezing trends of water in both the positive and negative pressure regimes, to which we provided evidence that it does. The present question from the reviewer is now addressing the linear approximation that we have put forward in Eq. (1) our paper. The reviewer notes that the approximation of Eq. (1) does not match experimental data in the positive pressure regime by Kanno et al. (1975). So, is this expression expected to perform better in the negative regime than in the positive pressure regime?

The short answer is, yes. This expression is most appropriate for pressures less than 500 atm, where the slope of the freezing line and the melting point line are approximately parallel. We have specified this more thoroughly in the revised manuscript. (See paragraph 6 of the Introduction)

First, we want to also mention that Eq. (1) is a first-order, linear, approximation to a curve. The approximation has been expanded around the reference pressure of 1 atm. Thus, the further away the pressure deviation dP from 1 atm, the less accurate the approximation will be.

But more importantly, we only expect this approximate form to be valid when the freezing point line and melting point line can be considered approximately parallel. As explained in the revised manuscript, experimental and simulation studies of water homogeneous freezing curves (including Kanno et al.) show that they become more parallel as the pressure approaches negative values. A recent simulation study by Montero de Hijes (2023) suggests that the freezing and melting lines should be parallel for pressures below 500 atm. We have cited this evidence in our revised manuscript.

With this limitation in mind, the predictions of Eq (1) are indeed consistent with the Kanno et al. data. Extending Eq. (1) to moderate positive pressures (e.g. 200 atm) provides excellent agreement with the experimental homogeneous freezing temperature depression measured by Kanno et al. Eq. (1) predicts a freezing point depression of 1.5K, which is exactly what is observed in the measured curve. (See the guidelines added onto the figure from Kanno et al. below).

Meanwhile, extending the approximation of Eq. (1) farther into the positive pressure regime (e.g. beyond 500 atm) exhibits a growing discrepancy between the predicted value and the experimental measurement. The equation predicts 3.65K depression, but the experimental result is 5-7K.

For the scope of this study, we feel that this limitation on Eq. 1 is justified. Our goal is to provide a simple framework to help quantify the effects of negative pressure on freezing temperatures. For atmospheric science applications, the pressure deviations dP will not extend drastically far from 1 atm, and thus a first-order approximation can be safely applied. This expression need not be broadly applicable at positive pressures because enhancement in freezing temperature is only expected when pressures/tension are negative.

[Figure]

☑ Lines 340–341: "Our findings provide additional perspectives to those of Lintunen et al. (2013), who showed a tendency for suppression of ice nucleation in the xylem of vascular plants". What is meant by this sentence?

**Incorporated into revision:** Yes

This result from Lintunen et al. (2013) seemingly contradicts our results, but the connections between this experiment in plant xylem and our simulations are not straightforward. We have

moved this reference to the section where we analyze our simulation results critically with experimental findings.

☑ Lines 342–343: what is meant here by "significant"? In the order of kPa or in the order of MPa? The order of magnitude is decisive for the impact negative pressure has on the ice nucleation rate and should be mentioned.

**Incorporated into revision:** Yes

Our estimates and evidence from examples in nature indicate that the highest magnitude of negative pressure we may see frequently in nature are around -500 atm. Although a few Kelvin may not seem significant, in the atmospheric context of mixed-phase clouds a few degrees can strongly influence cloud glaciation. In the range of -5 to -25 C, relevant to convective clouds, the abundance of active ice forming nuclei increases by approximately an order of magnitude for a 4-K change in temperature. We have added this discussion to both the introduction and discussion section of the revised manuscript.

**Reviewer #2 - Prof. Valeria Molinero**

This manuscript undertakes an investigation into the impact of negative pressure on the heterogeneous nucleation of ice. This study employs molecular simulations with two coarse-grained models of water, mW and ML-mW, in contact with model graphitic surfaces with varying hydrophilicities, yet comparable ice nucleation efficiencies. To this end, the authors present simulations of ice nucleation that apply negative pressures by two distinct means, namely a barostat on a system without a vapor phase and capillary pressure on a liquid-vapor system. The authors observe that the results are consistent within the uncertainty of the simulations. The findings of this study indicate that the pressure dependence of the heterogeneous nucleation temperature at a given nucleation rate is almost the same as that for the homogenous temperature in the same model. The authors explain this coincidence in the slopes through classical nucleation theory, assuming that the only property dependent on temperature and pressure is the chemical potential. In this regard, the interpretation of this study is based on water activity and employs approximations concerning the temperature and pressure dependence, which may not accurately elucidate the positive pressure side of the freezing line.

The topic addressed in this paper is of significant importance. However, I have observed that the conditions of the simulations are not consistently and adequately defined, and the discussion and conclusions contain unwarranted generalizations. Consequently, I am of the opinion that a revised manuscript could significantly enhance the presentation, analysis, and discussion, and result in an excellent paper. Thus, I recommend that the authors address the questions and issues raised in what follows. I have addressed them in the order they appear in the text.

Bianco et al. PRL **2021** cited in the manuscript

Dhabal D, Sankaranarayanan SK, Molinero V. *Stability and Metastability of Liquid water in a Machine-learned Coarse-grained Model with Short-range Interactions*. The Journal of Physical Chemistry B. **2022** 126(47):9881-92].

Evans LF. *Two-dimensional nucleation of ice*. Nature. **1967** Jan 28;213(5074):384-5;

Evans LF. *Ice nucleation under pressure and in salt solution*. Transactions of the Faraday Society. **1967**;63:3060-71].

Kanno, H., Speedy, R.J. and Angell, C.A., **1975**. *Supercooling of water to-92 C under pressure*. Science, 189(4206), pp.880-881)

Lu, J., Chakravarty, C. and Molinero, V., **2016**. *Relationship between the line of density anomaly and the lines of melting, crystallization, cavitation, and liquid spinodal in coarse-grained water models.* The Journal of Chemical Physics, 144(23), p.234507

Qiu Y, Lupi L, Molinero V. *Is water at the graphite interface vapor-like or ice-like?*. The Journal of Physical Chemistry B. **2018** Jan 3;122(13):3626-34.

**Introduction Section**

☑ I would like to bring to the attention of the authors that the use of the term "density anomaly" by them to describe the negative slope of the melting line, dTm/dp) coexistence, is not appropriate. The density anomaly of water pertains to the non-monotonic relationship between the density of the liquid and temperature. Thus, it would be more appropriate for the authors to refer to dTm/dp)coexistence < 0 as the negative slope of the melting line.

**Incorporated into Revision:** Yes

We agree that there is vagueness in how the term is used in various fields, so it is not the most appropriate term for our purposes. We have replaced our use of the term to be unambiguously referring to the property of negative thermal expansion.

☑ Equation 1 shows that Thom and Tmelt are parallel, but it is important to acknowledge in the introduction and discussion of the paper that this is not generally the case in experiments or simulations. The narrow range of validity of equation 1 is evident in the steepest slope of Thom(p) compared to Tm(p) at positive pressures in both experiments (Kanno et al., 1975) and simulations for mW (Lu et al., 2016), anTIP4P/Ice (Bianco et al., 2021) and ML-BOP (Dhabal et al., 2022). The authors should clarify in the manuscript that equation 1 is not applicable across all pressure ranges. This is due to the approximations employed to derive equation 1, which should be explicitly discussed in the paper. Importantly, equation 1 assumes that pressure and temperature only impact the chemical potential. The approximation that the ice-liquid surface tension is independent of pressure is reasonable for negative pressures, but not valid at positive pressures (Montero de Hijes et al., 2023).

**Incorporated into Revision:** Yes. (Paragraph 6 of Introduction).

We do state in the original manuscript, shortly after introducing Equation 1:

"According to this approximation, the slope of $\Delta T / \Delta P$ is parallel to the liquid-solid phase coexistence line, given by the Clapeyron equation. Detailed studies support that the slope of homogeneous freezing lines is not actually parallel to the melting point line (Bianco et al., 2021; Espinosa et al., 2016), but Rosky et al. (2022) has shown that, as an approximation, it holds true for pressures ranging from 1 atm to -1000 atm."

In the revised manuscript, we have added the additional references mentioned by the reviewer and significantly strengthened this section. Specifically, we have included the study by Montero de Hijes et al.(2023) into our discussion of Eq 1. This feedback from the reviewer has greatly improved the specificity and justification of the pressure range where $T_{hom}$ and $T_{melt}$ can be approximated as parallel.

☑ It should be noted that even the heat of fusion and the change in volume upon melting are dependent on pressure (Dhabal et al., 2022). The authors may alleviate the issues they encounter in Section 3.1, where the predicted and computed slopes of deltaT/deltap do not match, by incorporating this dependence.

The authors acknowledge that the heat of fusion and change in volume are pressure and temperature dependent. These dependencies are neglected when deriving Eq. (1) (See appendix C of Rosky et al. 2022), with the goal of establishing a simple framework for estimating the magnitude of freezing temperature enhancement expected from negative pressure in water. For atmospheric and experimental applications, the pressure and temperature deviations dP and dT will not extend drastically far from 1 atm, and thus this type of first-order approximation can be justified.

In the revised manuscript, we have added a sentence to the Introduction section clearly stating that the latent heat, ice-liquid surface tension, and molar volume difference are approximated as constant with changes in temperature and pressure relevant to this study.

Regarding the disagreement between the predicted and computed slope for the ML-mW heterogeneous freezing line:

We have extended our discussion of the possible physical sources of this observation as a result of the reviewer's insightful comments throughout this review. In this case, we are not sure that including these dependencies would help explain the steepened slope of the heterogeneous freezing line. The homogeneous freezing lines show great agreement with the predicted slope without accounting for the pressure and temperature dependence of the heat of fusion and volume change. Thus, incorporating these dependencies in the heterogeneous case should similarly have minimal impact on the predicted slope.

(See also our response to the reviewer's later comments related to the dependence of latent heat and molar volume on pressure, and the extent to which this influences the interpretation of our results.)

☑ The authors appear to be unaware that the melting and homogeneous nucleation lines of mW at pressures ranging from under -2000 atm to over 10000 atm were previously reported and discussed in Lu et al. (2016) "*Relationship between the line of density anomaly and the lines of melting, crystallization, cavitation, and liquid spinodal in coarse-grained water models*." The results for mW demonstrate that upon increasing pressure, the slope of Thom is steeper than that of Tm - consistent with the experimental results of Kanno et al (1975). The paper by Lu et al. should be cited when referencing previous simulations of the pressure dependence of freezing and melting in simulations and mW in particular. Additionally, the authors should be aware that equation 1 is not valid for all pressures.

**Incorporated into Revision:** Yes

We thank the reviewer for bringing our attention to this study. The results of Lu et al. (2016) provide yet another example of data suggesting that the melting point line and freezing line are roughly parallel in the negative pressure regime. It is definitely important to cite this study and we have done so in the first paragraph of Methods, as well as in the Introduction section. This and the reviewer's previous comment have both contributed to a more thorough discussion of the scope of Equation 1 in our manuscript.

☑ In line 68, the contact angle of mW water on the graphite surface of this study is reported as 86°. It may be more appropriate to refer to these surfaces as "ice nucleating" rather than as "hydrophilic substrates." If the hydrophilicity of the substrate plays a role in obtaining negative pressures in the capillary configuration, it is unclear why the authors chose a surface with a contact angle of almost 90°.

**Incorporated into Revision:** Yes

We have replaced the term hydrophilic with ice-nucleating throughout the manuscript. We chose to conduct heterogeneous simulations (substrate immersed in bulk water) with the mW-graphite system in order to compare our results against literature using the same interaction potential. For the water capillary bridges, we did not employ the mW model and only used the ML-mW model which has a smaller contact angle with the substrate.

☑ It should be noted that the correct name of the ML-mW model contains a dash between "ML" and "mW," and this should be corrected throughout the text.

**Incorporated into Revision:** Yes

**Methods Section**

The methods section of the manuscript lacks important information and is difficult to follow. The following points should be addressed:

**Response:** In addition to addressing the following points, we have re-organized the Methods section to be more clear and easy to follow.

☑ i. In Figure 1, the authors label (a) as "unconfined" and (b) as "confined." However, since both scenarios are periodic, they appear to be the same slab of liquid in contact with IN surfaces on the two boundaries of the slab. The authors should clarify how they handle the periodic boundary conditions for these cells to explain why they are different.

**Incorporated into Revision:** Yes (Paragraph 4 of Methods).

Addressed in detail in a later response (Please see the response in Results, Discussion, Conclusions section). Even though they both have PBC, the "unconfined" water has a vertical dimension nearly twice that of "confined" water.

☑ ii. The pressure for the supercooling referred to in line 83 and the cooling simulations should be explicitly stated.

**Incorporated into Revision:** Yes

☑ iii. The procedure to identify ice using OP seems to be the same as in Rosky et al. 2022, where about a third of the water molecules are identified as ice before crystallization. However, this data is not provided in the current manuscript, and the authors should clarify the identification process and provide the data.

**Incorporated into Revision:** Yes.

We have now specified the procedure used to identify the nominal freezing temperature in the revised Methods section.

The authors will add to the database repository for this manuscript: Files for each simulation containing the data [Timestep,  time(ns), temperature, N_ice, N_water, ice/water ratio] for each simulation run.

☑ vii. The manuscript is confusing and inaccurate in describing the ensemble of the simulations and the way pressure and temperature are controlled. The manuscript states that the simulations are done in the NPH ensemble with a Nose-Hoover thermostat to make it NPT, but the input file indicates that the thermostat used is the one from the canonical ensemble by velocity rescaling of Bussi et al. 2007, and the barostat is Berendsen's for equilibration and then Nose-Hoover for the collection run:

fix        2 water nve

```
fix        3 water temp/csvr ${TEMP} ${TEMP} 500.0 ${SEED}
fix        4 water press/berendsen iso ${PRES} ${PRES} 1000.0 modulus 20000

fix        2 water nph iso ${PRES} ${PRES} ${PCOUPL}
fix        3 water temp/csvr ${TEMP} ${END_TEMP} ${TCOUPL} ${SEED}
```

The authors should correctly describe the ensemble (NPT), and what are the actual thermostat and barostat they used and their damping constants. The use of isotropic control of the pressure may bias the growth of ice they are using to determine the freezing temperature and result in the relatively large dispersion of the freezing temperatures observed in this work (it may also result in hindrance of complete crystallization at low temperatures, as seen in Rosky et al **2022**).

**Incorporated into Revision:** Yes

We have now stated correctly in the paper: For equilibration, we use Berendsen barostat for pressure and Bussi stat (canonical velocity rescaling) for thermal stat. For production runs, we use Nose-Hoover for barostat and Bussi for thermal stat. We have also included the damping constants which are 5 and 10 picoseconds for temperature and pressure, respectively.

Regarding isotropic control of pressure, we are not aware of evidence suggesting isotropic control may be related to freezing temperature, particularly since we are only interested in the onset of crystallization, or nucleation, which should be minimally affected by the (an)isotropy of pressure. We understand that our method of using a sigmoidal fit to the ice/water ratio implies that the growth of ice is also included in the measure of freezing temperature. We think that this effect is probably not significant in our results due to the small box size and is accounted for in the freezing temperature uncertainty bounds. Particularly for the heterogeneous ice freezing simulations, the temperatures are warmer and thus the ice growth is quicker, making this even less of a concern.

As addressed in response to the reviewer's previous comment, we agree that it is necessary to clarify the methodology used in identifying freezing temperature, and this has been elaborated on in the revised manuscript.

☑ viii. What cooling rate was used in the simulations? Was it the same for both models?

**Incorporated into Revision:** Yes

Yes, the same cooling rate of 0.25 K/ns is used for all simulations in this study. This clarification has been added to the Methods section (first paragraph).

☑ x. Could the authors please explain how they achieved the same water-substrate area, given that it depends on the contact angle of water on the surface and the height of the water capillary? Did they tune the height of the cell or add/remove water molecules?

**Incorporated into Revision:** Yes

This is achieved by iteratively adding onto the y-axis box dimension until the correct surface area is found (within 6% of target value). The y-axis dimension of the cell is tuned. This detail has been added to the Methods.

☑ xii. Lines 134-137 indicate that squares and diamonds will be used for the two types of configurations, but Figure 2 only shows circles. What type of simulation cells were used for Figure 2? This information is not included in the caption.

**Incorporated into Revision:** Yes

We have updated the figure caption to specify which simulation cell is being used to produce the data in the figure.

The manuscript already states that "data points will be presented using circles to indicate unconfined heterogeneous freezing in the configuration shown by Fig. 1(a)." We believe no further revision is needed to the main text.

☑ xiv. Line 143 states again that the carbon of Lupi et al. is hydrophilic to mW, when it is essentially neutral (contact angle 86, which is not indicated until line 166). The contact angle should be indicated in this line 143 so that readers can judge how hydrophilic it is.

**Incorporated into Revision:** Yes

This has been added. Note we also revised "hydrophilic" to be "ice-nucleating" throughout the manuscript.

☑ Why was the carbon surface tuned to have a much lower contact angle of ~50o with ML-mW?

**Incorporated into Revision:** Yes

We wanted the ML-mW ice-nucleating substrate to exhibit the same magnitude of freezing temperature enhancement over the homogeneous freezing temperature as does the mW graphite substrate. Thus we tuned the ML-mW substrate potential to match this. In other words, we tuned the substrate so that $(T_{het} - T_{hom})$ at 1 atm is roughly the same for both mW and ML-mW. In the revised manuscript we elaborate more on the fact that the two water–substrate systems have different contact angles and discuss how this could be relevant to interpreting our results.

☑ iv. The procedure to determine the freezing temperature from the q6(T) of Rosky et al. 2022 corresponds to the nucleation and growth, not just nucleation. The authors should clarify this because the growth rate of ice is expected to decrease with extension, as the liquid is more tetrahedral.

**Incorporated into manuscript:** Yes. (First paragraph of Methods).

Since our approach of identifying the freezing temperature uses the inflection point as the nominal freezing point, the reviewer makes a good point that if the growth rate is slower, this will bias the nominal freezing temperature towards cooler temperatures. This effect is probably not significant in our results due to the small box size and natural variability in freezing temperature (uncertainty bounds). Particularly for the heterogeneous ice freezing simulations, the temperatures are warmer and thus the ice growth rate is quicker, making this even less of a concern. The authors do agree that this point is certainly worth taking into consideration. In addressing a previous comment from this reviewer on clarifying the methodology of identifying freezing temperature, the process of identifying the freezing point has been elaborated on in the revised manuscript. (See first paragraph of Methods section)

> ☑ v .The authors use the range of the interaction potential to explain the small size of the cell, but they should also consider the length scale of the structural correlations in the liquid. These correlations decay in about 1 nm, suggesting that water in cells with a liquid column of 2 nm or less is dominated by interfacial phenomena and should not be a good representation of larger systems. The authors should discuss the implications of these small sizes and consider adding a simulation of a larger system.

**Incorporated into manuscript:** Yes. (Paragraph 4 of Methods).

Our ''unconfined'' simulation cell dimension of 5 nm is at least 2x greater than the correlation length, and there is a region (~ 3 nm) where water can be considered nearly unaware of substrate. Finite size is always there but in terms of its impact on heterogeneous ice nucleation, it should be okay. The authors did some tests of these previously and this seemed to be the case. (Note that our definition of "unconfined" is explained in our response to the reviewer's later comment).

To provide an explicit response to this reviewer's concern, we doubled the z-dimension of the simulation cell and repeated the direct MD simulations of heterogeneous freezing at 1 atm to check against size effect. As we hoped, the larger box size produced the same nucleation rate on the substrate, using 20 cooling ramps.

In the revised manuscript we now address the structural correlation decay length and explain that our cell size is at least 2x larger than this length scale.

[Figure]

| Substrate separation | Temperature of J_het = 10^24 (s-1 m-2) | Temperature bounds |
|---|---|---|
| 16 A | 234.1538 | 3.7/2 |
| 24 A | 226.6225 | 3.8/2 |
| 30 A | 227.59 | 3.5/2 |
| 50 A ("unconfined") | 227.8 | 2 |
| 100 A (shown on left) | 227.3 | |

☑ vi Regarding the change in slope of the melting line, Lu et al. JPC 2016 show that for mW cavitation is reached before the extension that makes the liquid less dense than ice.

**Incorporated into manuscript:** Yes. (Final paragraph of Discussion).

The phenomenon of increased ice nucleation rate due to dynamic agitation or surface-related activity in experiments that we are studying is indeed limited in feasibility by the requirements that (1) ice is less dense than liquid, (2) the magnitude of tension can be realized in cloud droplets (3) Freezing is favorable over cavitation (or can cavitation precede freezing? - open question!)

Because of requirement 1, we specifically want to stay away from the region of phase diagram where liquid becomes less dense than ice.

Regarding point 2, although this is another open question, existing measurements of Laplace pressure in nature seem to extend to hundreds of atmospheres, but rarely to extreme negative pressures where we might encounter the delta_v inflection point.

Regarding point 3, some experiments indicate that cavitation may be linked to ice-nucleation, but this is an area of great uncertainty. The authors agree that, in the absence of finite size effects, the competition between cavitation and freezing, and whether or not cavitation can precede freezing, are fascinating and relevant questions that must be addressed to fully understand the potential of negative pressure to influence heterogeneous freezing in atmospheric cloud droplets. We have included this topic in our extended discussion on key questions of interest for future research.

☑ ix. Line 97 states that "the substrate molecules are held fixed with zero velocity". However, the authors should be aware that when the cell expands isotropically at negative pressure, the substrate will expand concomitantly (there are no forces keeping the substrate atoms together), resulting in a change of the ice nucleation properties of the surface. To avoid these issues, the authors should ensure that the graphite surface is rigid, not fixed (although LAMMPS does not like to evolve rigid periodic bodies!) or that they change the pressure by removing water molecules at constant volume rather than changing the dimensions of the cell.

**Incorporated into manuscript:** Yes.

Our simulation cell is only allowed to expand along the z-axis. Therefore there is no lateral expansion, so no change in substrate lattice geometry. As long as the z-dimension is large enough to not induce confinement (5 nm, which is the case here), then there should be minimal impact on nucleation. We have clarified in the Methods sections that the x,y-dimensions of the simulation cell are fixed.

☑ xi. Why was an ice cluster size of 25 molecules selected for sampling their positions? Is this the expected size of the critical nuclei at the conditions of the simulations?

The ice embryos are identified manually by selecting the earliest cluster of ice-like molecules meeting the conditions that:

1 - It can be clearly identified visually by eye.

2 - The cluster size grows with time beyond that point, indicating that it has reached critical radius.

The actual critical nuclei size was not taken into consideration. 25 molecules just happened to be the typical size where the above conditions would be met, but would range from 14 to 60 molecules.

Here is one example of what the initial ice cluster might look like during the manual visual identification process. (When doing the manual identification, only the ice-like molecules are shown. The substrate and edges of the capillary bridge are hidden to help remove unconscious bias) The molecules are colored by cluster.

[Figure]

☑ xiii. Can the authors explain why the uncertainties in That are much larger for ML-mW than for mW?

The larger uncertainty comes from the fact that for the same number of freezing events, the ML-mW freezing events take place over a wider range of temperatures than the mW freezing events do. The authors speculate that the uncertainty seems to increase with the slope (absolute) of the iso-rate or melting curve. However, we do not have a plausible rationale for why this happens.

**Results, Discussion, Conclusions**

☑ Line 174 "larger substrate area … would decrease the observed intensive heterogeneous nucleation rate" is probably wrong, as the nucleation rate is already normalized by the area.

**Incorporated into Revision:** Yes.

We understand the way that this sentence is written in the original manuscript could be confusing to readers. The authors mean to explain that the intensive nucleation rate (#/s/area) that we are able to sample within the timeframe of our simulations is dependent on the substrate surface area and the cooling rate.

Thus, the cooling rate and substrate area can be thought of as the two levers in our simulation set-up that adjust the range of nucleation rate coefficients available for sampling (and the corresponding temperature range of those nucleation rate coefficients).

The feedback makes it apparent that the way we have formulated this sentence in the manuscript is confusing to readers. We have revised the wording of this section to communicate more clearly.

☑ Lines 183-185 "Most significantly, we observe that the increase in temperature as a function of pressure for jhet is linear to within the sampling uncertainty, indicating that the use of a linear approximation for $\Delta T / \Delta P$ is appropriate for heterogeneous ice nucleation." This conclusion is not in agreement with experiments of pressure dependence of heterogeneous nucleation. For example, it has been shown that the pressure dependence of ice nucleation on many potent organic crystals IN is milder than for the melting line, resulting in a merging of the melting and freezing line at high pressures [Evans 1967]. The authors should refrain from generalizing about pressure dependence of heterogeneous and homogeneous nucleation from the small range of pressures and nucleating surfaces covered in their simulations. It would be more appropriate to discuss which factors may explain that the heterogeneous nucleation line is parallel to the homogeneous and melting lines at negative pressures in the simulations, and to which extent it can be expected that these results hold for ice nucleation with other substrates.

**Incorporated into manuscript:** Yes. (Paragraph 5 and 6 of Section 3.1).

We agree with the reviewer's assessment. Simulating different substrates and studying what differences, if any, result in the slope of constant heterogeneous nucleation rate coefficient lines would be a very worthwhile study to take this research topic to the next stage.

We have incorporated this perspective into our discussion of the $(\Delta T / \Delta P)_{het}$ slope and have also included the Evans 1967 reference.

☑ Line 189, again improper use of "water density anomaly" replace by "negative slope of the melting line" or "higher density of the liquid respect to ice"

**Incorporated into manuscript:** Yes.

Addressed in previous comment.

☑ Lines 212-222, the argument that the values of Tm, enthalpy of fusion and change in molar volume are ambiguous for heterogeneous nucleation does not make sense to me. In the framework of CNT used in this manuscript, these values are those of the bulk phase and the role of the interface becomes apparent only in the surface tensions and their temperature and pressure dependence. Those derivatives, dg/dT and dg/dp are the ones that the authors should focus on when addressing why the analyses that assume them to be zero do not provide a quatitative agreement with the data. The temperature dependence of the various surface tensions involved in the nucleation of ice at the mW-graphite interface have been discussed in Qiu et al. JPC B 2018 wonder whether the differences they see for mW and ML-mW are not rooted on the difference in hydrophilicity of the carbon-like surfaces used in the two sets of simulations. The results and discussion in this Qiu and Molinero 2018 paper may help address that issue.

**Incorporated into manuscript:** Yes.

For clarification, when we say that the values of $T_m$, enthalpy of fusion, and change in molar volume are ambiguous for heterogeneous nucleation, we mean that those values may be different for water near the substrate compared to the values in the bulk liquid. As such, the quantitative details of those differences are, at this point, elusive. This speculation seems to be consistent with Qiu 2018 et al. (2018), where free energy, enthalpy, and entropy of the water at the interface is studied in comparison to the bulk liquid.

We appreciate this reviewer's insightful comments and have incorporated this perspective into our analysis. The discussion in Qiu et al. JPC B 2018 is indeed very helpful in shedding light on this topic. We have incorporated this into our analysis in the following ways:

- We have added a paragraph discussing the surface tension between water and substrate as a potential factor in the slope of the $dT/dP_{het}$ line. (See paragraph 6 of Section 3.1).
- The influence that different substrates (and the surface tension between the water, ice, and substrate) may have on the quantitative slope of the heterogeneous freezing line in P–T coordinates is noted as a topic requiring further investigation. (See final paragraph of Discussion).
- The findings by Qiu et al. 2018 that "liquid water at the graphite interface is not ice-like or vapor-like: it has thermodynamics similar to that in the bulk liquid." seems to lend support to our speculation that the mW model heterogeneous and homogeneous freezing lines are parallel due to the thermodynamic properties at the graphite-mW interface being similar to that of bulk water. (Added to paragraph 5 of Section 3.1).

☑ Last paragraph of section 3.1, regarding the changes in enthalpy of fusion and difference in molar volume, it is important that the authors first perform the correction of these values with p and T [in experiment, as well as in ML-BOP and TIP4P/2005 these quantities have considerable slope, see figure 3 of Dhabal et al. JPC B **2022** op cit above], and if that does not explain the results consider the change in the surface free energies, which is where all the effect of the surfaces is in the formalism they adopted.

We thank the reviewer for the insightful comment. The changes in enthalpy of fusion and molar volume as a function of pressure and temperature are explicitly neglected in the derivation of Eq. (1). (See Appendix C of Rosky et al. 2022). We have also clarified this in the revised manuscript.

Since latent heat and volume difference both vary with pressure, one might include a pressure dependence (or correction) in Eq. (1) to obtain a more precise expression. However, these corrections are very small, and can thus be neglected for the purposes of this study.

For example, if we add a correction term to Eq. (1) to account for the change in liquid density using values extrapolated from Figure 3 of Dhabal et al. (2022), we accrue only a fraction of a Kelvin adjustment in the predicted freezing temperature at -1000 atm. (This correction can be added by keeping the $\frac{1}{2}(v_l(p) - v_l(p_0))$ term in Eqn C.4 of Rosky et al. 2022 Appendix).

Because Eq. (1) is a first-order approximation to a curve, it can be expanded around any reference pressure in the regime where valid (below 500 atm). In this study we use values taken at 1 atm. However, one might choose to expand around -500 atm instead, using the $T_{melt}$, $l_f$, and $\Delta v_{ls}$ at that pressure. This will produce a slightly different slope estimate that should provide a good match to the data in the vicinity of -500 atm. (This is not convenient for real water however, since those values are not easily measured).

Here we show, using -500 atm as an example, that using the pressure-adjusted values for $T_{melt}$, $l_f$, and $\Delta v_{ls}$ in Eq. (1) produces only a small change in the (dT/dP) slope.

- The Appendix of Rosky et al. (2022) provides a table of $\Delta v_{ls}$ and $T_{melt}$ values for the ML-mW models at 1 atm, -500 atm, and -1000 atm. From this table we can take $\Delta v_{ls}$ = -1.17 cm$^3$/ mol and $T_{melt}$ = 295 K at -500 atm.
- Extrapolating from the ML-BOP data in Figure 3 of Dhabal et al. (2022), the latent heat release $l_f$ at -500 atm might be approximated as 5857.6 + 167 J/mol.

Using all these values, the new slope at -500 atm is 0.057 K/MPa. This new slope is just 10% different from the slope at 1 atm. This amount of difference would not be distinguishable within the uncertainty bounds of the simulated (dT/dP)$_{het}$ data.

Regarding the reviewer's latter suggestion of considering the change in surface free energies: We feel that this is an important consideration and have incorporated it into our discussion of the ML-mW heterogeneous freezing curve. As we don't believe we can add any conclusive insights

on this without simulating different substrates, we have highlighted this topic in our discussion as a key question for future work. The perspectives from the reviewer have been very constructive and helpful to the authors.

☑ Line 235, eq 3 and eq 1 do not consider heterogeneous ice nucleation, why would their combination in eq. 4 account for heterogeneous nucleation?

While Equation 1 does not explicitly show the relationship to heterogeneous nucleation, it is applicable because it is based on the dependence of the water activity on pressure and temperature. Since water activity can be used to predict nucleation rates, both homogeneous (Koop) and heterogeneous (Knopf), heterogeneous nucleation is embedded within Eq 1. We also note that the focus of the first part of the manuscript is an effort to explicitly verify that Eq 1 does indeed apply to heterogeneous nucleation.

Equation 3 is a way to reformulate the delta P term in Eq 1 into variables that we can control or measure.

In essence, Equation 4 is the change in the water activity with pressure or temperature cast into variables that we can address in some experimental systems.

(See also our response to the reviewers comment on water activity).

☑ Line 238 "given the previous conclusion that terms sigma_lv and theta do not change significantly with pressure": that was not a conclusion but an assumption, because there was not data presented for either of these quantities as a function of pressure. The sentence must be edited to reflect that it is not a conclusion but an assumption or inference. It is known, however, that the surface tensions change with temperature [Qiu et al. JPC B **2018** presents data for mW] and the authors could account for that instead of using values at 298 K.

**Incorporated into Revision:** Yes

We agree with the reviewer and have revised this sentence accordingly.

In regards to accounting for the change in surface tension with pressure presented by Qiu et al. (2018):

Our main reason for refraining from exploring such correction terms in this manuscript is because the scope of this paper is focused on proposing a simple framework that can help atmospheric scientists and experimentalists start to quantify the effects of negative pressure on freezing temperatures. The approximation put forth in our Eq. (1) is a first-order approximation to a curve, expanded around the reference pressure of 1 atm. For atmospheric applications, the pressure and temperature deviations dP and dT will not extend drastically far from 1 atm, and thus a first-order approximation can be applied. Since the approximation is expanded around 1 atm, experimentally measured values of water can be used and this framework does not require

investigators to input precise thermodynamic values for water in other regions of the phase diagram. We agree that extending the expression for the shape of heterogeneous freezing curves across a broad range of temperatures and pressures is an exciting area of research for future investigation.

☑ The first paragraph of section 3.3: when preparing the water-filled cells with height 1.8, 2.4 and 3 nm, what are the pressures at which they are evolved? The Laplace pressure that you deduce for the same height capillaries in the previous simulations? This is not clear in the text, please add detail.

**Incorporated into Revision:** Yes

They are evolved at the 1, -500, and -1000 atm pressure settings. Since we know the freezing temperature of an "unconfined" box of water at those pressures, we can see if the 1.8, 2.4, and 3 nm confinement has any influence on the freezing point.

☑ Line 266: "unconfined" configuration does not seem to be less confined than the other slabs – please explain clearly where is the lack of confinement in that cell; I do not see it.

**Incorporated into Revision:** Yes

We define "unconfined" as the point where the separation between the substrate surfaces has no effect on the freezing behavior of the water, and where the substrate surfaces are separated far enough to leave a bulk-like volume of water between which is uninfluenced by either substrate layer. In the methods section, we have revised the manuscript to clearly define how we use the term "unconfined" and keep this term in quotations throughout the paper to acknowledge that we do not mean it in a literal sense.

Our simulation cell dimensions for the "unconfined" configuration is about 5x5x5 nm. There isn't a clear cut, but 5 nm separation seems to be a sufficient distance for dipping into the "bulk" region of water. This has been used in many studies. Just to name a few:

1. Lupi, L., Hudait, A. & Molinero, V. Heterogeneous Nucleation of Ice on Carbon Surfaces. J Am Chem Soc 136, 3156–3164 (2014).

2. Cox, S. J., Kathmann, S. M., Slater, B., Slater, B. & Michaelides, A. Molecular simulations of heterogeneous ice nucleation. I. Controlling ice nucleation through surface hydrophilicity. J. Chem. Phys. 142, 184704 (2015).

3. Bi, Y., Cabriolu, R. & Li, T. Heterogeneous Ice Nucleation Controlled by the Coupling of Surface Crystallinity and Surface Hydrophilicity. J Phys Chem C 120, 1507–1514 (2016).

Of course, finite size effects may always come into play, but in terms of impact on ice nucleation behavior this separation distance seems to be sufficient to avoid confinement effects. One way

to rationalize this is that water density oscillation decays to virtually zero beyond 1.5 nm, regardless of hydrophilicity. See

> 1. Fig. 2 in Cox, S. J., Kathmann, S. M., Slater, B. & Michaelides, A. Molecular simulations of heterogeneous ice nucleation. II. Peeling back the layers. J. Chem. Phys. 142, 184705 (2015).

> 2. Fig. 2 in Bi, Y., Cabriolu, R. & Li, T. Heterogeneous Ice Nucleation Controlled by the Coupling of Surface Crystallinity and Surface Hydrophilicity. J Phys Chem C 120, 1507–1514 (2016).

Essentially, when the confinement distance is beyond 1.5x2=3 nm, there is very little interaction of density oscillations from both confining surfaces.

To verify all this, we have run additional simulations of the "unconfined" cell with the z-dimension doubled to verify that the ice nucleation rate on the substrate is unchanged. These results are included in our response to the reviewer's related remark earlier in this document, and have been mentioned in the revised manuscript.

> ☑ I understand the increase in ice nucleation for the 1.8 nm water slab cell is due to concurrent help of the nucleation surfaces, as explained in Hussain, Sarwar, and Amir Haji-Akbari. "*Role of nanoscale interfacial proximity in contact freezing in water.*" Journal of the American Chemical Society 143.5 (**2021**): 2272-2284. I recommend the authors to cite that manuscript if that is the phenomenon they are observing.

**Incorporated into Revision:** Yes

This study by Hussain, Sarwar, and Amir Haji-Akbari aims to address the same experimental phenomenon that motivates this work and is thus highly relevant to consider and cite. In their study study, they are looking to see if confinement of water between IN and the air-water interface will increase the nucleation rate to possibly account for the efficiency of contact-nucleation, whereas our hypothesis focuses on the possibility of increase in nucleation rate due to Laplace pressure.

In our study, our aim is to avoid confinement effects in order to isolate the influence of Laplace Pressure. The details of the confinement phenomenon are tangential to the goals of our research study. Nevertheless, the connections between our findings and those of Hussain, Sarwar, and Amir Haji-Akbari (2021), are very interesting and have been cited in the revised manuscript.

> ☑ End of section 3.3 "Other research (Elliott, 2021) supports that capillary theory can extend to the nano-scale used in our simulations, which our results corroborate. Meanwhile, our results are also consistent with Almeida et al. (2021), who indicate that the capillary theory breaks down with separations less than ≈ 20" . I do not see where you corroborate the validity of capillary theory to the nanoscale nor where do you show that capillary theory breaks down below 2 nm (ice nucleation in confinement does not

defy capillary theory). Please make explicit your evidences and arguments in this discussion.

**Incorporated into Revision:** Yes

The authors agree with the reviewer's logic here since capillary theory in Almeida (2021) deals with the variation of water contact angle with separation distance, whereas we are dealing with whether confinement affects ice nucleation. There does not appear to be an obvious connection here. We have removed this argument from the manuscript.

☑ Lines 284-5: aren't the capillaries too small to conclude that there is a statistically significance preference for ice nucleation between 2 and 2.5 nm from the air-water interface? Please clarify what are the sizes of the capillaries you analyze to reach that conclusion, as I do not find it justified. They seem to be influenced by finite size effects.

**Incorporated into Revision:** Yes

In response to this comment, we simulated a wider capillary bridge and did not see the spatial preference for ice nucleation reproduced. We have revised our conclusions accordingly.

The data used to obtain the distribution shown in Figure 5(d) are from the 24-A, and 30-A tall capillaries, which both have the same width of 60-A (60-A separation between the air-water interfaces). The dimensions of these capillaries are shown in Figure 1(c). The distribution plot includes a total of 65 ice-nucleation events and the uncertainty bars are appropriately calculated using Poisson uncertainty. We note that there does seem to be a tendency for ice to nucleate with higher probability between 20-A and 25-A compared to the 25-A to 30-A bin. However, follow-up simulations prompted by this comment show that this is not observed in a wider capillary where the air-water interfaces are separated by 120-A. Thus, we have revised our conclusions to reflect this and included these new results in the Appendix of the revised manuscript.

As detailed in the Appendix B: We prepared a capillary bridge that is 120-A wide and simulated 50 ice nucleation events. We are not able to directly compare the nucleation rate with the previously studied capillaries because the substrate-water surface area is different, but we can look at the ice-nucleation locations to see if the wider capillary bridge still shows a preference for 2 to 2.5 nm from the air-water interface. The results do not show a pronounced preference for ice-nucleation in the sub-surface region.

☑ In discussing whether ice forms or not at the air-water interface, the authors may want to take into account that premelting of water at the ice-vapor interface rules out the existence of heterogeneous nucleation of ice at the water-vapor interface (because premelting and heterogeneous nucleation require the opposite sign of ice binding free energies, as discussed in Qiu Y, Molinero V. Why is it so difficult to identify the onset of ice premelting?. The journal of physical chemistry letters. **2018** Aug 27;9(17):5179-82.

**Incorporated into revision:** Yes

**Response:** We agree that the presence of the premelting layer at the ice–vapor interface is relevant to understanding our results. We have pointed out the connection between the premelting layer and the lack of nucleation at the air-water interface and have cited this reference.

☑ In the conclusions section, make sure you do not generalize your results as a pressure parameterization beyond the regime that you measure, and I suggest incorporating into the discussion the role of the change in surface tension with pressure (at least the one for water-ice, reported recently for TIP4P/2005 by Montero de Hijes et al. JCP **2023)**, as otherwise you run against an experimental body of evidence that shows that the temperature of heterogeneous nucleation is not necessarily parallel to either the melting or homogeneous nucleation lines [see the papers by Evans cited above].

**Incorporated into revision:** Yes. (Paragraph 6 of Discussion).

We have incorporated the investigation by Montero de Hijes et al. JCP 2023 into our discussion. In general, we have revised the manuscript to ensure the scope of these results are more explicitly limited to the negative pressure regime between 1 atm and -1000 atm. This is appropriate for our research objectives because ice nucleation enhancement due to pressure is only theorized in the negative pressure regime.

The question of how much the heterogeneous freezing slope can be influenced by different substrates is an open question for further refinement. We have cited the Evans 1967 study and critically analyze our results in the context of experimental findings.

☑ A central finding in this study is that the heterogeneous nucleation temperature is rather insensitive to pressure in the range of negative pressures (probably because the surface tension is also relatively insensitive to pressure in this range).
☑ The changes in freezing temperature upon extension seem quite modest to me. Considering that water at negative pressure is doubly metastable with respect to ice and vapor, then to which extent the extension of supercooled water is able to promote nucleation in time scales that are short compared to cavitation?

**Incorporated into revision:** Yes

Our estimates and evidence from examples in nature indicate that the highest magnitude of negative pressure we may see frequently in nature are around -500 atm. Although a few Kelvin may not seem significant, in the atmospheric context of mixed-phase clouds a few degrees can strongly influence cloud glaciation. In the range of -5 to -25 C, which is relevant to convective clouds, the abundance of active ice forming nuclei increases by approximately an order of magnitude due to a 4-K change in temperature. We have discussed and quantified this in the revised manuscript. The topic of cavitation and its links to ice nucleation is a fascinating one that we believe is best suited to be expanded upon in future work. We have included this topic in the extended discussion.

☑ Lines 345-346 "Conversely, imposing isochoric conditions has been shown to greatly increase the stability of supercooled water so that it can be used for cryopreservation (Powell-Palm et al., 2020) – That paper refers to the stability with respect to cavitation, not crystallization. I do not see the relevance of this sentence regarding stability against cavitation in the context of what is being discussed in that paragraph… but if you think it is important, clarify that it refers to stability with respect to cavitation.

Our understanding after reviewing the paper Powell-Palm et al., 2020 is that it does, indeed, refer to the stability of supercooled water through the suppression of ice nucleation via density fluctuations. There is a link to cavitation which we refer to in the revised discussion.

☑ Finally, the model that the authors use for the analysis is based on the dependence of the excess chemical potential of water with respect to ice as a function of pressure and temperature, expressed in the equations of CNT. That has similarities to the water activity approaches, such as the one of Knopf and Alpert cited in the manuscript. It would be important the the authors elaborate on the connection of theirs and Knopf and Alpert approach, adn whether they are equivalent.

**Incorporated into Revision:** Yes. (Paragraph 5 of Discussion).

The original manuscript includes a discussion on water activity and to that we have added some additional points. Here, we further elaborate on the topic of water activity:

The paper by Koop et al. (2000) shows that the same effect in homogeneous nucleation can be achieved by increasing the solute concentration or by increasing the pressure. Both of these can be attributed to change in the water activity. One of the major breakthroughs in that paper was to collapse the solute curves on each other. Solutes do affect the water activity differently. For example, sodium chloride behaves differently than lithium chloride despite the fact that both have a van't Hoff factor of 2 at infinite dilution.

Daniel Knopf and Peter Alpert (2013) extended this concept with the activity-based immersion freezing model (ABIFM), which applies the same idea to heterogeneous freezing. Koop showed that the nucleation rate curve would follow the melting line curve, only that the nucleation rate curve is shifted by some constant value of the water activity. The work of Knopf and Alpert (2013) indicates that heterogeneous freezing will follow a similar trend. This is also demonstrated by Archuleta et al. (2005), as well as Cantrell and Robinson (2006).

These activity-based approaches are consistent with our results, so we expect that the insights gained through our molecular dynamics simulations can contribute to the implications of the pressure-dependence of activity.

Koop, T., Luo, B., Tsias, A. *et al.* Water activity as the determinant for homogeneous ice nucleation in aqueous solutions. *Nature* **406**, 611–614 (2000).

Archuleta, C. M., DeMott, P. J., and Kreidenweis, S. M.: Ice nucleation by surrogates for atmospheric mineral dust and mineral dust/sulfate particles at cirrus temperatures, Atmos. Chem. Phys., 5, 2617–2634 (2005).

Cantrell, W.; Robinson, C. Heterogeneous freezing of ammonium sulfate and sodium chloride solutions by long chain alcohols. Geophys. Res. Lett. 2006, 33 (7), L07802.

Knopf, Daniel A, and Peter A Alpert. "A water activity based model of heterogeneous ice nucleation kinetics for freezing of water and aqueous solution droplets." *Faraday discussions* vol. 165 (2013): 513-34.

---

## Author Response (AR2)

Dear Dr. Tim Garrett,

On behalf of myself and my co-authors, I would like to thank the editor and reviewers for the valuable feedback on our revised manuscript.

We agree that the abstract should be shortened and are glad for the opportunity to make the suggested improvements for the final version. The revised abstract is now close to the 250 word limit and makes a stronger statement on the potential implications for ice nucleation in the atmosphere. We would like to confirm with the editor that our statements are reasonable for this publication (not too provocative)?

The additional comments from the reviewers have each been addressed and our responses are below in blue text. We greatly appreciate the attention to detail from the reviewers, which has ensured that the final manuscript is of a good quality.

Kind Regards,
Elise Rosky

**Reviewer #1**

The authors have addressed the comments of the reviewers carefully and in detail. There are only few points left that need to be corrected or clarified.

Line 60: "The molar volume difference between water and ice is negative…" The way it is formulated, it remains ambiguous whether the difference "water – ice" or "ice – water" is negative. The formulation should be improved to remove this ambiguity.

> Thank you for this suggestion. The improvement has been made near Line 60 where we now express the explicit definition $\Delta v_{ls} = v_l - v_s$.

Line 247: "…dependent, the…" instead of "…dependent; The…"

> Done.

Line 274–275: "…from Qiu et al. (2018) which identified mW water at the mW-Carbon interface as having thermodynamics similar to that of the bulk liquid." This sentence is grammatically incorrect. Should "having" be removed?

> Done.

Line 303: "We find that the data follows a linear trend as anticipated." Consider a more cautious formulation: "We find that the data can be described by a linear trend as anticipated".

> Done.

Figure 4: Fonts are in general rather small and the one chosen for the x-axis in panel b is much too small and must be increased. The label below "1 atm reference value" on the x-axis of panel b is not readable.

> Fonts have been increased.

Line 401: the abbreviation for "ice nucleating particle" is usually "INP" and not just "IN".

> This is true. We also use the term "ice nuclei" in this paragraph, which can be abbreviated as "IN". In the equations discussed in this paragraph we use the abbreviation "IN" in the subscripts. In this case it's a little more convenient to stick with "IN" in order to not muddle the subscripts which also use "P" to denote pressure. We hope this abbreviation is still clear enough to the readers.

Lines 432–434: "Insofar as our results depend on the water activity (Nemec, 2013), they are consistent with previous efforts to cast nucleation in that framework (Koop et al., 2000; Knopf and Alpert, 2013).": Note that water activity and pressure both act on the chemical potential difference between ice and the water phase, but they are not the same or interchangeable.

> The authors agree with this comment. We have adjusted the text accordingly to clarify that pressure and water activity are related through their contribution to the chemical potential difference between liquid and ice.

Lines 443–445: "Experimental measurements of freezing inside porous material observed no freezing enhancement in pores with confinement on similar size scales as studied here (~2 nm diameter) (e.g., Marcolli, 2014),…": Note that the experiments compiled in Marcolli (2014) are slurry experiments. The pores have no air-water interface. Therefore, no Laplace pressure is expected. Moreover, ice nucleation does not occur within the pores but ice grows into the pores when the temperature is low enough. A valid reference should be given for this statement.

> The presence of Laplace pressure is not necessary in this argument because this paragraph is considering how different confinement geometries will result in mixed influences on the ice nucleation rate. One may anticipate an increase due to Laplace pressure, but this could be suppressed or further enhanced if confinement effects are present.

> Figure 3 of Marcolli 2014 shows freezing point depression in completely filled pores with increasingly small diameters. This is a good summary of experimental results showing that small scale confinement does not always result in enhancement in nucleation rate.

In the final manuscript we keep the same reference but adjust the text so that there is no longer a discrepancy between our statement and the provided reference. The revised section can be found in the second-to-last paragraph of Discussion section.

*Marcolli, C.: Deposition nucleation viewed as homogeneous or immersion freezing in pores and cavities, Atmospheric Chemistry and Physics, 14, 2071–2104, 2014.*

Lines 448–449: "The key mechanism for rate enhancement in the slit pore configuration is the density oscillations induced by a flat interface (Cox et al., 2015; Bi et al., 2016; Lupi et al., 2014), which is destroyed by curved geometry." Please explain what you mean with this sentence. Up to here, the enhancement in pores has been explained by the Laplace pressure arising in pores with air-water interface. Density oscillations are a completely new explanation.

The reviewer makes a good point. We now introduce this concept at an earlier point in the manuscript, when we first describe the confinement effect in the 18A tall simulation cell (Section 3.3 Paragraph 1).

**Reviewer #2**

I commend the authors on the greatly improved manuscript that has resulted from their thorough revisions. The authors have addressed all my questions, comments and concerns. I have only one comment/suggestion and a suggestion/request of correction that I hope the authors will address in the final version of the manuscript:

Line 272) "This could indicate the thermodynamic properties of mW water are less influenced near the substrate compared to the ML-mW model. This interpretation is supported by evidence from Qiu et al. (2018) which identified mW water at the mW-Carbon interface as having thermodynamics similar to that of the bulk liquid."

My comment/suggestion: Note that the reason why the thermodynamics of mW at the carbon surface is almost same as for bulk water is because the contact angle for the liquid on that surface is almost 90 degrees. The authors could use the equations of that paper to derive how a contact angle of 50 degrees for the surface with ML-mW impacts the thermodynamics, and whether that explains the differences they see for these two water-surface models.

> We will incorporate this important point into the text. We thank the reviewer for the suggestion and look forward to exploring this idea in more detail in future work.

Line 356) "This lack of heterogeneous ice nucleation in the immediate vicinity of the air–water interface could be related to premelting at the ice–vapor interface, described for the mW model by Qiu and Molinero (2018)."

My comment/request: Qiu and Molinero 2018 demonstrates that heterogeneous ice nucleation cannot occur on a surface that exhibits premelting presented using solely thermodynamics and nucleation theory. It is a general derivation that does not depend on results for any particular model. Please replace "described for the mW model " by "as deduced using thermodynamics and nucleation theory"

> Done.

---

## Author Response (AR3)

Dear Dr. Tim Garrett,

We appreciate the helpful suggestions from the editor. We now provide a revised abstract incorporating all of your comments, which we feel has resulted in significant improvements.

Given the additional opportunity to make edits for the final version, a few other minor improvements have been made to the body of the manuscript. First, subsection headers have been added to the Methods section, making this section more organized and easy to navigate. Secondly, the Concluding Remarks section has been trimmed and polished in order to be more consistent with the revised abstract. Minor grammatical changes were also made.

We would like to thank the editor and reviewers again for the time and effort they have put into providing constructive feedback on this manuscript.

Sincerely,
Elise Rosky